

**Throughfall exclusion and fertilization effects on tropical dry forest tree plan-**
**tations, a large-scale experiment**
German Vargas G.[1,2,*], Daniel Perez-Aviles[3], Nannette Raczka[4], Damaris Pereira-Arias[3], Julián
Tijerín-Triviño[5], L. David Pereira-Arias[3], David Medvigy[6], Bonnie G. Waring[7], Ember Morrisey[8],
Edward Brzostek[4] and Jennifer S. Powers[1,3]
[1]Department of Plant and Microbial Biology, University of Minnesota, St. Paul, MN USA 55108.
[2]School of Biological Sciences, The University of Utah, Salt Lake City, UT 84112.
[3]Department of Ecology, Evolution, & Behavior, University of Minnesota, St. Paul, MN USA 55108.
[4]Department of Biology, West Virginia University, Morgantown, WV USA 26506.
[5]Department. de Ciencias de la Vida, Grupo de Ecología Forestal y Restauración, Universidad de Alcalá, Madrid,
España 28801.
[6]Department of Biological Sciences, University of Notre Dame, Notre Dame, IN USA 46556.
[7]Grantham Institute on Climate Change and the Environment, Imperial College London, London UK, SW7 2AZ.
[8]Division of Plant and Soil Sciences, West Virginia University, Morgantown, WV USA 26506.
*Correspondence to*: German Vargas G. (gevargu@gmail.com)
Keywords: biomass allocation, diameter growth, leaf area index, litterfall, nodule production, primary productivity,
root productivity, soil microbial biomass.





**Abstract.** Across tropical ecosystems, global environmental change is causing drier climatic conditions and increases in nutrient depositions. Such changes represent large uncertainties due to unknown interactions between drought and nutrient availability in controlling ecosystem net primary productivity (NPP). Using a large-scale manipulative experiment, we studied whether nutrient availability affects the responses of three component NPP fluxes (stem growth, fine roots production, and litterfall) to through-fall exclusion in 30-year-old unmanaged mixed plantations of six tree species native to the tropical dry forest of Costa Rica. We used a factorial design with four treatments: control (CN), fertilization (F), drought (D), and drought+fertilization (D+F). While we found that a 13-15% reduction in soil moisture only led to modest effects in the studied ecosystem processes, NPP increased as a function of F and D+F. At the same time, NPP increases with nutrient additions were larger in the plots without throughfall exclusion. The relative contribution of each biomass flux to NPP varied depending on the treatment, with woody biomass being more important for F and root biomass for D+F and D. Moreover, seasonal canopy cover was maintained longer in the fertilized plots. Belowground processes such as nodulation and microbial carbon use efficiency (CUE) also responded to experimental treatments, with a decrease in nodulation for F plots and an increase in CUE for F and D plots. Species functional type (i.e., N-fixation or deciduousness) and not the experimental manipulations were the main source of variation in tree relative growth rates. Our results emphasize that nutrient availability moderately constrains ecosystem processes in tropical dry forests, but this depends on water availability.



## 1 Introduction

Global environmental change is affecting primary productivity in tropical forest ecosystems. Among the main factors behind this variation in productivity are the changes in the hydrologic regime due to an increase in rainfall seasonality (Feng et al., 2013), increases in atmospheric water demand (McDowell et al., 2020), and regional decreases in soil moisture (Seneviratne et al., 2010). In other words, the tropics are getting drier. Results from observational studies in wet and tropical dry forests (TDFs) found that droughts may increase tree mortality rates (Powers et al., 2020), reduce above-ground biomass productivity (Phillips et al., 2009; Castro et al., 2018), reduce the production of seeds and flowers (O'Brien et al., 2018), and increase the abundance of high wood density and deciduous tree species (Swenson et al., 2020; Aguirre-Gutiérrez et al., 2020; González-M et al., 2021). Collectively, this evidence suggests that these ecosystems are changing in response to ongoing climatic variation. However, how tropical forests respond to drought depends on other environmental factors, such as soil fertility, history of disturbance, and fire regime, among others (Chazdon et al., 2005; Brando et al., 2014; Brodribb et al., 2020; Becknell et al., 2021; Wu et al., 2022). Accounting for how these environmental variables modulate ecosystem responses to drought will enhance our understanding of the impacts of global environmental change on the carbon cycling (Bonan, 2008), particularly in tropical forests, which play a disproportionate role in global carbon dynamics and provide ecosystem services to a quarter of the world's population (Wright, 2005; Lewis, 2006).

One largely overlooked factor is the potential role of nutrient availability in mediating tropical forests' vulnerability to drought. Tropical forests exist in a highly heterogeneous mosaic of soil fertility, parent material, and topography (Townsend et al., 2008; Augusto et al., 2017; Waring et al., 2021), properties that shape forest composition (Condit et al., 2013; Werden et al., 2018a), and function (Clark and Clark, 2000). Nutrient-limited environments harbor a greater proportion of slow-growing, drought-tolerant species, whereas fast-growing drought-avoiding species dominate nutrient-rich environments (González-M et al., 2021; Oliveira et al., 2021), which suggests that soils play an important role in determining the heterogeneity of tropical forest ecosystem responses to drought. Moreover, anthropogenic activities can cause an increase in atmospheric nitrogen and phosphorus deposition across ecosystems (Wang et al., 2017), and yet the consequences of these changes in combination with rainfall variation remain unknown in tropical forests (Matson et al., 1999; Hietz et al., 2011). In summary, the empirical evidence needed to characterize drought-nutrient interactions or the effects of soil characteristics on drought responses has yet to be documented but is highlighted as a priority to parameterize vegetation dynamics models (Smith et al., 2014).

### 1.1 Nutrient and water availability affect primary productivity

In principle, high nutrient availability could alleviate drought stress as plants with higher leaf nitrogen maximize water use efficiency at the cost of photosynthetic nitrogen use efficiency (Lambers et al., 2008). For example, nutrient limitation negatively affects water use efficiency in crop species and tropical seedlings (Santiago, 2015). Soil fertility also affects the responses of forest productivity to rainfall variation, such that TDF stands growing in more fertile soils tend to show higher increases in productivity with higher rainfall than stands in nutrient-poor soils (Medvigy et al., 2019; Becknell et al., 2021).



Other processes besides primary productivity provide insight into ecosystem responses to global
environmental change. Leaves, and more precisely canopy cover, are the main center for carbon assimilation in
forest ecosystems. Recent evidence has shown that the patterns of leaf flushing and leaf shedding are changing at the
global scale as a consequence of climate change (Piao et al., 2019). While it is well documented that tropical leaf
phenological cycles depend on plant water status and the start of the rainy season (Frankie et al., 1974; Borchert,
1994), little is known about how soil fertility interacts with water availability to affect leaf production in TDFs. A
decrease in leaf canopy cover affects productivity by decreasing the photosynthetic area (Doughty and Goulden,
2008), while changes in the timing of leaf flushing/shedding may create a cascade of effects with unknown
consequences on other ecosystem processes, which will affect organisms that depend on these processes (Coley,
1998). Moreover, changes in the timing of leaf production could make plant species more susceptible to the presence
of herbivores in greater abundances (Janzen, 1981; Neves et al., 2014). Thus, quantifying the effects of rainfall
reductions on leaf production is key to disentangling the interactions between soil moisture and soil fertility, and to
understanding the coupling among primary productivity, canopy processes, and climate.

The extent to which the interactions between nutrient and water availability affect below-ground processes
is highly uncertain, particularly in TDFs (Phillips et al., 2016; Allen et al., 2017). Moreover, the increase of specific
nutrients, *i.e.*, via nitrogen deposition, might also cause an imbalance in stoichiometry or increase water demand
with unknown consequences for tropical forests as plants will adjust to compensate by increasing transpiration rates
or producing more root biomass (Lu et al., 2018; Waring et al., 2019). This evidence suggests that the effects of
drought on ecosystem productivity could vary depending on the edaphic conditions and atmospheric deposition of
nutrients. At the heart of this uncertainty is the degree to which soil microbial carbon use efficiency (CUE; the
proportion of total carbon consumed that is used to grow new biomass) either acclimates or shifts in response to
changes in water and nutrient availability. Theory suggests that microbes with high CUE produce more biomass that
upon death becomes protected from future microbial attack by adhering to mineral surfaces (Cotrufo et al., 2013).
Under drought, the CUE of the microbial community may decrease owing to the need to use carbon for survival
strategies rather than for growth (Schimel et al., 2007). When rainfall returns to these soils, this shift toward a low
CUE microbial community may drive significant soil carbon losses since available soil carbon will be metabolized
by a less efficient microbial community. However, there is the potential that reducing microbial nutrient limitation
may alleviate the impacts of drought on CUE due to microbes investing less energy in resource acquisition for
protective molecule production (Schimel et al., 2007). Given the potential for shifts in nutrient and precipitation
regimes to alter microbial CUE, identifying the extent to which these drivers alter CUE in TDFs is critical to
increasing our predictive understanding of soil carbon cycling in this important biome (Knorr et al., 2005; Chadwick
et al., 2016).
**1.2 Experimental framework**

Contrary to tropical rain forests, carbon cycling in TDFs is likely limited by both water and nutrient
availability (Lugo and Murphy, 1986; Castro et al., 2018; Medvigy et al., 2019; Becknell et al., 2021). This co-
limitation of resources highlights the importance of quantifying the individual and interactive roles of these two



factors in shaping ecosystem processes in this important and threatened biome (Hoekstra et al., 2005; Miles et al.,
2006). Large-scale manipulative experiments are needed to understand the interactions of drought and nutrient
limitation, although to date an experiment testing these two factors simultaneously has not been implemented in
tropical forests. While nutrient addition experiments have shown mixed (positive, weak and none) effects on tree
growth in tropical forests (Wright et al., 2018; Hou et al., 2020), results from throughfall exclusion experiments
suggest an initial decrease in woody productivity over the first two years and an increase in mortality after five years
(Meir et al., 2015). Most of these large-scale experiments have been conducted in wet tropical forests (mean annual
rainfall > 2000 mm) (Meir et al., 2015; Wright et al., 2018), even though TDFs seem to be equally vulnerable to
drought (Powers et al., 2020), and once accounted for up to 40% of tropical forest area (Murphy and Lugo, 1986).
To understand how both water and nutrient availability controls important primary productivity fluxes that
contribute to carbon cycling, we established a large-scale, fully factorial experiment in mixed-species plantations as
model TDF stands. We used rain-out shelters covering 50% of the forest floor area to reduce soil moisture and/or
fertilizer applications to increase soil nutrient availability to investigate four questions. First, how do tree species
differ in their growth responses to throughfall reduction and/or fertilization? Second, do experimental manipulations
affect phenological patterns of leaf production? Third, what is the role of nutrient availability and/or changes in soil
moisture in controlling root production, nodulation and soil microbial CUE? Fourth, to what extent does nutrient
addition affect the responses of primary productivity to experimental drought? We predicted that species-level
growth and ecosystem-scale primary productivity would be negatively affected by reduced throughfall, but that this
effect would be less in plots that were also fertilized.

## 2 Methods

Our experiment was conducted for four years (2016-2020) at Estación Experimental Forestal
Horizontes (hereafter Horizontes), in northwestern Costa Rica (10.711°N, 85.578°W) (Figure 1). Before Horizontes
was incorporated into Área de Conservación Guanacaste (ACG), the lands were used for grazing and crops (Werden
et al., 2018b). Since 1989, Horizontes has served as a large-scale ecological and forestry research laboratory, and the
~7500 ha area encompasses a mosaic of TDF at different successional stages (0-80 years), 64 ha of timber
plantations trials of native TDF species (Gutiérrez-Leitón, 2018), restoration trials (Werden et al., 2020), a
Mesoamerican TDF arboretum (http://www.arbnet.org/morton-register/arboretum-del-bosque-seco-tropical), as well
as seed orchards of endangered precious wood species (M. Gutiérrez-Leitón *personal communication*). During the
study period total annual rainfall averaged ~1547 mm, with annual totals as follows: 1439 mm (2016), 2201 mm
(2017), 992 mm (2018), 919 mm (2019), and 2186 mm (2020), and median temperatures were 26.5 ± 1.6 °C during
the dry season and 25.6 ± 1.5 °C during the wet season (Fig. S1). Precipitation values were in range with the
historical average of ~1500 mm (Vargas G. et al., 2015). In Horizontes the start of the wet season is defined when
cumulative rainfall reaches 100 mm, which usually occurs in May and defines the beginning of the hydrological
year (*i.e.*, 12 months after cumulative rainfall reaches 100 mm) (Aragão et al., 2007; Waring et al., 2019). Therefore
our temporal scale is the hydrological year as in TDFs the start of the rainy season determines the beginning of leaf
production, seed germination, and other ecological processes (Murphy and Lugo, 1986).



## 2.1 Experimental design

We conducted our experiment in tree plantations that were established in 1991 (Gutiérrez-Leitón,
2018). The plantations consist of three 8-10 ha blocks that each contain one of three focal species combined with
one of four species from a pool of 11 species native to Northwestern Costa Rica (Fig. S2). The plantations have not
received any management (fertilization, liana cutting, thinning, etc.) for 25 years prior to our experiment, trees were
planted at a spacing of 3 x 3 m, and the understory now contains grasses, forbs, and a diverse community of 15
lianas and 50 trees/shrubs (Figure 1). We selected six species from the 11 that were planted (Table 1), which
represent functional types common to the TDF given the species ability to fix atmospheric nitrogen and their leaf
habit phenology (Xu et al., 2016; Powers and Tiffin, 2010). We took a tree-centered approach in locating the plots to
include at least six individuals of each focal species in the four treatments, with a minimum of 12 trees per plot. For
this reason, the plot area ranged from 120-360 $m^2$ and contained a two-species combination that we designated as
stand (Table S1). This experimental design was a compromise that allowed us to have at least four individuals of
each species within plots. Before selecting the plot locations, we did extensive surveys of tree diameters to ensure
that there were no systematic differences in tree diameters within species among treatments (Fig. S3). Soil samples
(0-10 cm depth) were collected in 2016 and 2021, by taking 7 to 10 cores (2.5 cm diameter) and compositing cores
by the plot. Particle size distribution was collected in 2016 (Table S2), extractable elements (Olson extractable Cu,
Zn, Mn, Fe, K, and P), and total C and N for samples collected in the fifth year (2021).

The experiment consisted of a fully factorial design with four experimental treatments: fertilization (F),
drought (D), drought+fertilization (D+F), and un-manipulated control (CN). We established four plot replicates, per
experimental treatment that each contained two of the six tree species, such that every species was represented in
one plot of each treatment. The D and D+F treatments consisted of a series of structures that covered 50% of the
surface area at each plot and were suspended at a 40° angle at distances from 0.4 to 2.5 meters above the ground
(Figure 1). The throughfall exclusion shelters were built with locally sourced materials including transparent
polycarbonate corrugated sheets, wood poles, and polyvinyl chloride pipes. To avoid lateral fine root growth outside
the throughfall exclusion structures, we dug a 50 cm deep trench around each exclusion plot that was covered with a
barrier of double folded 0.075 mm thick polyethylene film and then backfilled. Precipitation was routed off the
throughfall exclusion plots by a system of gutters and ground channels (Figure 1). For the F and D+F treatments, a
slow-release complete formula (macro- and micro-nutrients) nutrient fertilizer was broadcasted uniformly over the
entire plot area in two applications during the rainy season each year. From 2016 to 2018 we used Basacote® Plus
*3M* (Compo Expert GmbH), and then due to low market availability from 2018 through 2020 we used Osmocote®
Plus (The Scotts Company LLC) (Table S3). Nutrient addition rates were targeted to 150 kg N ha$^{-1}$ yr$^{-1}$ (Table S1),
similar to other large-scale tropical forest fertilization experiments (Wright et al., 2011; Alvarez-Clare et al., 2013;
Waring et al., 2019). Finally, because litterfall accumulated on the surfaces of the plastic panels, every two weeks
we used long brooms to sweep the litter off the panels and then place it under each panel.

## 2.2 Soil moisture





In each plot we quantified volumetric soil moisture at a 30 min frequency for the duration of the
experiment with an EM50 Digital data logger equipped with four 10-HS soil moisture probes (METER Group, Inc.
USA). Probes were distributed in two opposite pairs from the center of the plot, each pair consisting of a probe at 10
cm depth and another probe at 40 cm depth.

**2.3. Aboveground processes**

**2.3.1. Tree growth**

From 2016 to 2020, we measured the diameter at breast height (DBH) for all stems greater than 2.5 cm
DBH annually at the end of the growing season. These measurements included the plantation trees and every stem
that recruited into the 2.5 DBH size class before and during the experiment. All trees and shrubs were identified to
species level and classified into nitrogen fixation and leaf habit (evergreen or deciduous) functional types. For stems
between 2.5 cm and 10 cm DBH, diameter was measured with a diameter tape at a marked point 130 cm above the
ground. In the case of stems > 10 cm DBH, we measured DBH increments using band dendrometers set at 130 cm.
For each stem we calculated relative growth (RGR) as $RGR_{i\text{-}f} = log(DBH_f/DBH_i)/((DC_f\text{-}DC_i)/365)$, where $DC$
represents the day of the century, $i$ and $f$ final $DBH$ and $DC$ values for a given stem (Wright et al., 2011).

**2.3.2. Canopy productivity**

We measured canopy productivity using two complementary methods: litterfall traps and leaf area index.
To measure litterfall production, we deployed three 0.25 m$^2$ traps ~ 0.4 m above the ground in a transect along the
center of each plot. In plots with throughfall exclusion structures, traps were in the spaces between the
polycarbonate sheets. Litter was collected monthly from each trap, dried for 72 hours at 60 °C, and sorted into
leaves, small branches, flowers, fruits, and frass, and then weighed separately. We then calculated the annual
litterfall productivity in kg m$^{-2}$ yr$^{-1}$ for total litterfall (leaves, small branches, flowers, and fruits), leaves, and
reproductive litterfall (flowers and fruits).
Leaf area index (LAI) was measured in seven points at each plot (four in each corner and three along the
center) every 10 to 30 days with an LAI-2200C Plant Canopy Analyzer (LI-COR Biosciences, Lincoln, NE, USA).
The variation in sampling frequency was caused by logistical constraints that wet seasons occasionally imposed on
our ability to reach the plots. Because of the high abundance of species from the Fabaceae family in the plots, LAI
measurements were performed after sunrise (between 0900 to 1100 h) given the associated nastic movements in
leaves after dawn and before dusk (Minorsky, 2019). For that reason, we took each measurement using a 45° angle
cap towards the center of the plot and performed scattering correction before and after each measurement cycle
throughout the entire experiment (LI-COR Biosciences, Lincoln, NE, USA). LAI data were subsequently estimated
from the first four gap fractions using the software application FIV-2200 (LI-COR Biosciences, Lincoln, NE, USA).
In 2017 tropical storm Nate, which impacted 85% of the Costa Rican territory (Quesada-Román et al., 2020), caused
a significant LAI decrease during the month of November (Fig. S4). For that reason, we dropped the measurements



of November and December during 2017 from all the analyses involving LAI data. From the LAI data, we extracted
leaf area duration (LAD, $m^2 \, m^{-2} \, d^{-1}$), which describes the temporal dynamics and leaf persistence in the canopy of
broad-leaf plant communities (Ewert and Pleijel, 1999; Norby et al., 2003). LAD is defined as the area under the
non-linear curve of LAI as a function of the Julian day:
$$LAD = \int_b^e LAI\,(t)$$
Where, $b$, describes the beginning of the growing season in Julian days; $e$, the end of the growing season in
Julian days; LAI ($t$), the function of LAI temporal variation during the given growing season (Pokorný et al., 2008).
The growing season in this case is defined by leaf flushing and leaf fall (Norby et al., 2003). In our case, we
obtained the growing season parameters $b$ and $e$ from plot-specific and year-specific generalized additive models by
estimating the Julian days in which LAI starts to increase (positive slope change) from the minimum and when it
starts to decrease (negative slope change) after the maximum LAI (Methods S1). Then we fitted the LAI temporal
variation during the growing season and integrated it from $b$ to $e$ to obtain the area under the non-linear function
LAI ($t$) (Fig. S5). In addition to LAD, for each plot, we calculated the maximum LAI value during the growing
season ($LAI_{max}$, $m^2 \, m^{-2}$), minimum LAI during the dry season ($LAI_{min}$, $m^2 \, m^{-2}$), leaf-less period (LLP, d), the
beginning of leaf flushing (GSB, d) and the seasonal LAI enlargement (LAE, %) which is the percentage change in
LAI from the dry season to the wet season (Pokorný et al., 2008).
**2.4. Belowground processes**
**2.4.1 Fine-root and nodules production**
We measured fine root productivity from July 2016 through December 2020 using the ingrowth core
method (Waring et al., 2016). To do this, we installed seven ingrowth cores in each plot to a depth of 15 cm. The 8
cm diameter cylindrical ingrowth bags were made from 2 mm nylon mesh. The cores were collected two months
after deployment and a subsequent new set of cores was installed after collection. While deploying the cores, we
filled them with sieved, root-free soil collected on-site. During the first year of the experiment, cores were sampled
in the dry season. However, the clay-rich soils harden greatly during the dry season, which increased the difficulty
of deploying new bags during these times. Therefore, for three years of experimental ingrowth bags were harvested
in June, August, and November. After collecting the cores, fine roots were separated from the soil by washing them
over a 2 mm sieve. We counted the number of nodules on each root sample if present. Finally, root samples were
dried for 72 hours at 60 °C and weighed to estimate total fine root productivity in $kg \, m^{-2} \, yr^{-1}$.
**2.4.2 Microbial CUE and priming**
To analyze microbial CUE, we collected ten soil samples (5 cm diameter, 15 cm depth) from each plot
during the wet season in August 2019 and homogenized into one soil sample per plot. The samples were expedited
back to the University of Minnesota where a laboratory microcosm experiment was performed. Microbial CUE was
assessed using the $^{13}C$ glucose tracing method (Frey et al., 2013), briefly >97% $^{13}C$ glucose (Cambridge Isotope



Laboratories) at a rate of 400 ug C $g^{-1}$ soil was mixed with 25 g of each homogenized soil sample in 32 oz mason
jars (946.3 ml) with septa in the lids. Soil samples not rewet to maximum water-holding capacity (WHC) from D, F,
CN, and D+F plots were brought to 20% WHC with the addition of the glucose solution, with lab replications
yielding n=4. To examine the effects of rewetting, additional soil samples from each field treatment were rehydrated
with a glucose solution to maximum soil WHC (n=4). Additional control soils were incubated without the addition
of glucose and received the same amount of deionized water as non-rewet samples as a non-amended control,
bringing the total to 48 jar incubations. Soil microcosms were well mixed with water or substrate solution and
incubated for one week at room temperature. During this time the production of $^{13}CO_2$ and total $CO_2$ were assayed
every other day by taking gas samples from the microcosm headspace through the septa and inserting them into 12
ml Exetainer vials (Labco Limited). After gas samples were taken, jars were opened for ~ 20 minutes to allow for
gas exchange. After the experiment was complete, Exetainer vials were shipped to West Virginia University, where
each gas sample was measured using an LI-6400 (LI-COR Biosciences, Lincoln, NE, USA) and Picarro G2201-i
(Picarro Inc., Santa Clara, CA, USA). Glucose and soil organic matter-derived $CO_2 – C$ was calculated via mass
balance as described in Morrissey et al. (2017). Priming was then calculated as the difference in soil organic matter
$CO_2 – C$ between the microcosms that received substrate solution and those that received water. At the cessation of
the incubation, total microbial biomass was obtained by a chloroform fumigation method (Witt et al., 2000). Briefly,
8g of soil was suspended in 45 mL of 0.1 M $K_2SO_4$ with or without an additional 1ml of ethanol-free chloroform and
shaken for 4 hours (chloroform) or 2 hours (no chloroform) and filtered (90 mm GF/A filter paper). Extracts were
stored at -20°C until dissolved organic carbon was oxidized to $CO_2$ via persulfate digestion similar to Doyle et al.
(2004). Digestion efficiency was determined using a standard curve of yeast extract solution ranging from (0-200
mg C $L^{-1}$). The concentration and isotopic enrichment of the resulting $CO_2$ gas were measured on the Picarro. Total
and substrate-derived microbial biomass-C was calculated as the difference in C (mg) between chloroformed and
non-chloroformed soil extracts. Microbial CUE was calculated as substrate-derived biomass -C divided by the total
carbon consumed (substrate-derived $CO_2 – C$ and biomass -C) as described in Frey et al. (2013).
**2.5. Ecosystem productivity**
To quantify total net primary productivity (NPP) in kg $m^{-2}$ $yr^{-1}$, we summed total litterfall, wood, and
root production in each plot for a given year. To estimate wood production, we calculated stem aboveground
biomass (AGB) using allometric equations for tropical tree species (Chave et al., 2014). For the allometric
equations, we measured height of each stem using a Haglöf EC II-D electronic clinometer (Haglof Inc., Madison,
MS, USA) and obtained wood density data from a functional trait database for the TDF of Guanacaste (Powers and
Tiffin, 2010; Vargas G. et al., 2021). Mean wood density was substituted for species without wood density data.
Annual woody productivity then represented the sum of biomass increments from trees newly recruited into the 2.5
cm DBH size class plus biomass gain from increased diameters in planted and existing recruited trees. Additionally,
we calculated aboveground net primary productivity (ANPP) by summing only woody and litterfall productivity,
and the aboveground:belowground productivity ratios (AGB:BGB) by dividing ANPP by the root production in
each plot for each year.





**2.6. Statistical analysis**

To test whether the throughfall exclusion structures affected soil moisture, we performed a linear mixed-model with the change in soil moisture for a given plot as the response variable, the presence of the throughfall exclusion structure and the weekly timepoints from January 2017 to December 2020 were fixed effects, and probe nested within plot nested within stand as a random intercept. We ran separate models for each depth (10 and 40 cm), and for the wet season and dry season due to the strong rainfall seasonality. To obtain the change in soil moisture per plot, we divided the observation time into two periods, a pre-treatment (May 2016 to August 2016) that consisted of wet season soil moisture data before the shelters were set up, and an experimental period (January 2017 to December 2020). We did not consider the period between August and December 2016 as the soil was saturated when we established the rainout shelters. After removing outliers using the interquartile method, we calculated the median pre-treatment soil moisture ($SM_{PT}$, $m^3$ $m^{-3}$) for each probe in each plot. We then calculated the treatment effect as the percentage change between each soil moisture observation ($SM_i$, $m^3$ $m^{-3}$) and the $SM_{PT}$. This approach was considered given the high spatial heterogeneity in soil properties among plots (Table S2), making it impossible to compare values of volumetric water content. To investigate inter-annual variation in wet season soil moisture, we fitted an additional linear mixed model to test whether soil moisture in plots without throughfall exclusion varied as a function of year and depth, and probe nested within plot nested within stand as a random intercept. In both cases, we calculated type III sum squares and the F-value for each model and performed Tukey's honest significant difference test (Tukey's HSD) for multiple comparisons.

We tested the effects of the experimental treatments on aboveground and belowground processes by fitting a series (one for each response variable) of a two-factorial linear mixed effects model. In these models ecosystem processes were the response variables, the drought treatment was one factor, and the fertilizer treatment the second factor, we included their interaction, and the experimental unit (*e.g.*, litterfall basket) nested within plot, nested within stand as a random intercept. With this model, we were able to estimate the main effect of drought, the main effect of fertilization, and the interaction between drought and fertilization. We then calculated type III sum squares and the F-value for each model in an analysis of variance (ANOVA), with a Tukey's HSD for multiple comparisons. To analyze the response of CUE from soils that were held at field soil moisture to soils that were rewet, we calculated the natural log response ratio [*i.e.*, ln(RR)], defined here as the mean of the rewet soils CUE divided by the mean CUE of the field soil moisture soils. Values of ln(RR) 0 indicate a decline in CUE to rewetting. All data management, and statistical analysis were done using R software for statistical computing version 3.6.3 (R Core Team, 2021), and the packages mcvg (Wood, 2004, 2011), nlme (Pinheiro et al., 2019), car (Fox and Weisberg, 2019), and tidyverse (Wickham et al., 2019).

**3 Results**

**3.1. Soil moisture and fertilization**

At 40 cm depth, the change in soil moisture as a function of the pre-treatment period was larger in the plots with a throughfall exclusion structure (~ -13%) than in the plots without it (~ -4%) (Figure 2). At 10 cm we





observed an average change of -15% in throughfall exclusion plots, but this was not significantly different when
compared to the non-droughted plots (-9.43%) (Figure 2). Weekly median soil moisture values in the throughfall
exclusion plots oscillated between 0.21-0.42 $m^3$ $m^{-3}$ at 10 cm depth and 0.25-0.44 $m^3$ $m^{-3}$ at 40 cm depth, compared
to 0.22-0.43 $m^3$ $m^{-3}$ at 10 cm depth and 0.25-0.45 $m^3$ $m^{-3}$ at 40 cm depth for plots without throughfall exclusion
(Figure 2). Wet season soil moisture followed the inter-annual rainfall variability in which the average volumetric
water content was around 0.39 $m^3$ $m^{-3}$ during 2016 and 2017, while it was around 0.32 $m^3$ $m^{-3}$ from 2018 to 2020
(Fig. S6). At the end of four years, extractable soil P increased by 2-3 fold in plots receiving fertilizer, and
extractable Fe also increased (Fig. S7); however, none of the other soil chemical variables we measured differed
among treatments.

### 3.2. Above-ground responses

### 3.2.1. Tree diameter relative growth rates (RGR$_{dbh}$)

We found no evidence of changes in RGR$_{dbh}$ as a function of drought (D) and fertilizer additions (F) for
either understory (D: $F\text{-}v = 0.03$, $d.f. = 1$, $p\text{-}v = 0.8601$; F: $F\text{-}v = 0.22$, $d.f. = 1$, $p\text{-}v = 0.6580$) or plantation trees (D:
$F\text{-}v = 2.35$, $d.f. = 1$, $p\text{-}v = 0.1489$; F: $F\text{-}v = 1.14$, $d.f. = 1$, $p\text{-}v = 0.3041$). We found moderate evidence of an
interaction between drought and fertilizer for both plantation ($F\text{-}v = 5.16$, $d.f. = 1$, **$p\text{-}v = 0.0499$**) and understory
trees ($F\text{-}v = 5.04$, $d.f. = 1$, **$p\text{-}v = 0.0659$**) (Figure 3). Moreover, nitrogen-fixing functional group explained the
differences in RGR$_{dbh}$ for understory ($F\text{-}v = 21.11$, $d.f. = 1$, **$p\text{-}v < 0.0001$**) and plantation trees ($F\text{-}v = 4.18$, $d.f. = 1$,
**$p\text{-}v = 0.0512$**) (Figure 3) with non-N-fixers showing higher growth rates than N-fixers in both cases. On the other
hand, deciduous functional group showed weaker effects on RGR$_{dbh}$ for plantation trees ($F\text{-}v = 3.95$, $d.f. = 1$, **$p\text{-}v =$**
**0.0639**) (Table S4). In general, RGR$_{dbh}$ varied idiosyncratically among plantation species in response to the
experimental treatments, and in many cases RGR$_{dbh}$ in fertilized or drought+fertilizer plots was higher than in the
control or drought plots, but we did not find evidence of significant treatment effects (Fig. S8). We also found a
higher number of dead trees over the four years in plots with experimental manipulations and hence higher biomass
losses (Table S5). Mortality, recruitment, and survival for the trees and shrubs that were recruited in the plots did not
differ in response to the experimental treatments (Fig. S9).

### 3.2.2. Canopy productivity

The experimental manipulations showed no effects on fine litter production (drought: $F\text{-}v = 0.96$, $d.f. = 1$,
$p\text{-}v = 0.3473$; fertilizer: $F\text{-}v = 1.33$, $d.f. = 1$, $p\text{-}v = 0.2724$) and the production of leaves (drought: $F\text{-}v = 0.64$, $d.f. =$
$1$, $p\text{-}v = 0.4404$; fertilizer: $F\text{-}v = 1.39$, $d.f. = 1$, $p\text{-}v = 0.2646$). Nevertheless, the control plots produced on average
$0.69 \pm 0.14$ kg $m^{-2}$ of fine litter, which was 12 % lower than in the fertilized plots with $0.78 \pm 0.14$ kg $m^{-2}$, 13% less
than $0.79 \pm 0.24$ kg $m^{-2}$ of the drought plots, and 8% lower than $0.75 \pm 0.23$ kg $m^{-2}$ in drought+fertilizer plots. We
also found a 40% decrease in the production of flowers, seeds, and fruits with nutrient additions ($F\text{-}v = 4.84$, $d.f. =$
$1$, **$p\text{-}v = 0.0539$**) (Fig. S10), but no effects with the throughfall exclusion ($F\text{-}v = 1.54$, $d.f. = 1$, $p\text{-}v = 0.2449$). In all
the plots leaf area index (LAI) increased ~73% from the dry season (median LAI: 1.22) to the wet season (median



LAI: 5.10). All the metrics obtained from the LAI measurements showed some degree of change in response to the
fertilization treatment as these plots showed the highest maximum LAI ($LAI_{max}$), longest leaf area duration, shortest
leaf-less period, and on average leaf flushing started two weeks earlier than control plots (Fig. S11). However, these
changes where marginally significant only for $LAI_{max}$ ($F\text{-}v = 3.36$, $d.f. = 1$, ***p-v = 0.0928***).

### 3.3. Below-ground responses

### 3.3.1. Fine roots and nodule production

We found no evidence that differences in the production of fine roots were due to the throughfall
exclusions ($F\text{-}v = 0.25$, $d.f. = 1$, $p\text{-}v = 0.6227$) or nutrient additions ($F\text{-}v = 0.73$, $d.f. = 1$, $p\text{-}v = 0.4105$); despite that,
root productivity in the control plots ($0.112 \pm 0.06$ kg m$^{-2}$) was ~15% less than in the drought plots with $0.133 \pm$
$0.09$ kg m$^{-2}$, ~27% less than in the fertilized plots with $0.154 \pm 0.09$ kg m$^{-2}$, and ~24% less than in the
drought+fertilizer plots $0.149 \pm 0.12$ kg m$^{-2}$. In general, we observed a decrease in the production of nodules in the
fertilization treatment ($\chi^2 = 4.95$, $d.f. = 1$, ***p-v = 0.0262***), because only 1 nodule was observed in plots with nutrient
additions during the experimental manipulations. Interestingly, nodule production was the highest in plots with
drought and drought+fertilzer with 69 and 57 respectively, but we found little evidence this was different from 53
nodules counted in the control plots from 2016 to 2020 ($\chi^2 = 0.03$, $d.f. = 1$, $p\text{-}v = 0.8589$).

### 3.3.2. Microbial carbon use efficiency (CUE)

CUE was ~38% higher in soils from both the drought ($F\text{-}v = 4.31$, $d.f. = 1$, ***p-v = 0.0621***) and fertilized
plots ($F\text{-}v = 4.10$, $d.f. = 1$, ***p-v = 0.0678***) relative to control plots (Figure 4). When the soils were rewet in the lab,
the CUE exhibited a negative response as quantified by the ln(RR) for both the drought ($F\text{-}v = 5.66$, $d.f. = 1$, ***p-v =***
***0.0366***) and fertilization treatments ($F\text{-}v = 0.73$, $d.f. = 1$, ***p-v = 0.0809***) (Figure 4). There were interaction effects
between experimental treatments for both the CUE ($F\text{-}v = 5.33$, $d.f. = 1$, ***p-v = 0.0462***) and ln(RR) ($F\text{-}v = 4.76$, $d.f.$
$= 1$, ***p-v = 0.0597***), showing evidence of different responses to drought depending on nutrient availability and how
CUE was negatively affected by rewetting for drought plots (Figure 4). Soil priming was similarly influenced by
rewetting and across all treatments, the soils held at field soil moisture showed negative priming (Fig. S12).
Rewetting the soils in the lab led to greater soil C priming in the drought plots ($F\text{-}v = 5.33$, $d.f. = 1$, ***p-v = 0.0497***),
but not in the fertilized plots ($F\text{-}v = 0.0191$, $d.f. = 1$, $p\text{-}v = 0.8932$) (Fig. S12).

### 3.4. Ecosystem productivity and biomass allocation

Ecosystem level fluxes was more responsive to nutrient additions than to the throughfall exclusion (Figure
5). Net primary productivity (NPP) increased with nutrient additions ($F\text{-}v = 7.86$, $d.f. = 1$, ***p-v = 0.0178***), which led
to 17% and 19% higher NPP in fertilizer and drought+fertilizer plots respectively relative to the control plots
(Figure 5). Although we observed a 14% NPP increase in the drought plots ($F\text{-}v = 5.29$, $d.f. = 1$, ***p-v = 0.0431***), we
found no evidence this was different from the control plots (Figure 5). Consistently, when considering only above-



ground net primary productivity (ANPP) we found that fertilizer increased the amount biomass produced ($F\text{-}v =$
5.81, *d.f.* = 1, ***p-v* = 0.0362**) which was 15% and 19% higher for fertilizer and drought+fertilizer plots respectively
relative to the control plots (Figure 5). Moreover, the drought treatment decreased ANPP ($F\text{-}v = 4.58$, *d.f.* = 1, ***p-v* =**
**0.0575**). We found no evidence that the drought ($F\text{-}v = 0.30$, *d.f.* = 1, *p-v* = 0.5960) or fertilizer plots ($F\text{-}v = 0.35$,
*d.f.* = 1, *p-v* = 0.5645) allocated significantly more belowground biomass, despite the 13 and 15% reduction in the
aboveground to belowground ratio observed in the drought and fertilizer plots respectively relative to the control
plots (Figure 5). We did not observe interaction effects by the experimental treatments in either NPP ($F\text{-}v = 1.13$,
*d.f.* = 1, *p-v* = 0.30), ANPP ($F\text{-}v = 0.77$, *d.f.* = 1, *p-v* = 0.3991), or AGB:BGB ($F\text{-}v = 0.34$, *d.f.* = 1, *p-v* = 0.5695),
although the response to nutrient additions in the plots without throughfall exclusions was three and four times
higher for NPP and ANPP respectively relative to plots in the drought treatment (Figure 5, panel b).
**4 Discussion**

Here we present the first attempt to experimentally test whether the responses of primary productivity and

microbial carbon use efficiency to drought are limited by nutrient availability in a tropical ecosystem (Beier et al.,
2012). Our experiment is also the only large-scale rainfall manipulation study in the tropical dry forest (TDF) biome
(Meir et al., 2015). We found that a 13-15% reduction in soil moisture only leads to modest effects in the studied
ecosystem processes. By contrast, extractable P increased in the fertilized plots severalfold (Fig. S7) causing an
increase in primary productivity (both NPP and ANPP) (Figure 5), a decrease in the nodule production, a decrease in
the production of seeds and flowers (Fig. S10), increases in $LAI_{max}$ and LAD (Fig. S11), and an increase in CUE
when compared to the control plots (Figure 4). Variation in tree relative growth rates ($RGR_{dbh}$) were mostly due to
functional types rather than the experimental treatments. However, there was a significant interaction in how
understory trees responded to both treatments leading to a reduction in the differences between N-fixing and non-N-
fixing trees (Figure 3). Collectively, these results suggest that reducing soil moisture by a modest amount is not
sufficient to drive large reorganizations in the ecosystem, and that soil nutrient availability has a mild control over
short-term changes in TDF productivity. Below, we further explore the implications of these results in the context of
how soil fertility could affect tropical ecosystem responses to global environmental change.
**4.1. Nutrient and water limitations on ecosystem productivity**

In a broad sense, we found that nutrient availability had a stronger control on forest productivity than a

~15% reduction in soil moisture. While this result does not resonate with the expectation that water availability
imposes a greater limitation on productivity across environmental gradients than soil fertility (Harrington et al.,
1995; Santiago and Mulkey, 2005; Toledo et al., 2011; Sala et al., 2012; Poorter et al., 2016), it provokes the
question to what extent are tropical dry forests resilient to drought stress? Our data point to other aspects related to
drought intensity and not soil moisture alone that could be key factors in how water availability shapes TDF primary
productivity (Anderegg et al., 2013). Recent studies from northwestern Costa Rica have shown that abnormal
drought stress due to a strong ENSO event in 2015 caused biomass loss due to an increase in tree mortality, a



decrease in reproductive biomass production, and reductions in productivity (O'Brien et al., 2018; Castro et al.,
2018; Powers et al., 2020). The main characteristics of the 2015 ENSO were the elevated temperatures and a
substantial rainfall reduction for the region (Santoso et al., 2017), which can increase the severity of drought effects
in forest ecosystems due to increased atmospheric water demand (Brodribb et al., 2020; McDowell et al., 2020).
Thus, while throughfall exclusion experiments manipulate soil moisture, it is possible that a combination of factors
such as the vapor pressure deficit, the rainfall patterns (intensity and seasonality), and their linkages to soil moisture,
is a more important aspect of drought stress for forested ecosystems.

We observed the strongest experimental signal in the fertilization treatment (F and D + F) regardless of the

throughfall reductions. Such responses agree with known evidence of nutrient limitation on productivity in various
tropical forests (Alvarez-Clare et al., 2013; Wright et al., 2018; Waring et al., 2019), which has also been observed
in ecosystem models for the TDF (Medvigy et al., 2019). Interestingly the contribution of each biomass flux to NPP
depended on the combined effects of drought and fertilization, with root productivity contributing more to droughted
plots and woody productivity to fertilized plots (Fig. S13). Such changes in root and woody biomass production are
comparable to responses in secondary wet tropical forests to nutrient additions (Wright et al., 2018). In a nearby
secondary TDF Waring et al. (2019) found no significant effect of nitrogen and/or phosphorus additions on
productivity, however, in contrast to that study, our experiment included the additions of both macro and micro-
nutrients (Table S2). Moreover, the increase in productivity as a function of fertilization depended on the presence
of throughfall structures with non-drought showing the greatest increase (Figure 5, panel b). This confirms the
colimitation of water availability and soil fertility on TDF productivity, where forests in fertile soils are more
responsive to increases in rainfall than forests in infertile soils (Becknell et al., 2021). At the same time, our results
are comparable to other throughfall exclusion experiments in which fine litter production was not affected by the
drought treatment in a consistent manner (Nepstad et al., 2002; Brando et al., 2006; Schwendenmann et al., 2010),
with a lot of its variation possibly linked to climatic variability (Brando et al., 2008).
**4.2 Canopy dynamics and tree growth**

Canopy dynamics did not show strong variation in response to the experimental treatments The timing of

leaf flushing, period of no leaves, leaf area duration, and maximum canopy cover showed some mild responses to
the fertilization treatment, indicating that added nutrients may allow plants to retain canopy cover for longer periods
(Fig. S11). It is possible that the timing of leaf phenology may also depend on intra- and interspecific responses to
environmental factors that shape soil water availability including temperature, atmospheric water demand, and soil
water retention. For example, the tree species *Coussarea racemosa* A. Rich modified its vegetative and reproductive
phenology in response to a rainfall manipulation in the eastern Amazon (Brando et al., 2006), while at the forest
level changes were observed in $LAI_{max}$ but not the timing of leaf production (Brando et al., 2008). In a throughfall
exclusion experiment combined with fertilization in loblolly pine (*Pinus taeda* L.) plantation there were no changes
in the $LAI_{max}$ in response to rainfall reduction but an increase in the $LAI_{max}$ in the fertilized plots (Samuelson et al.,
2014), which is qualitatively consistent with our data.



No species showed significant changes in RGR$_{dbh}$, but the understory trees showed a reduction in the

differences between N-fixing and non-N-fixing trees. For F and D this was due to a reduction in growth rates by

non-N-fixing trees, while for D+F due to an increase in the growth rates by N-fixing trees (Figure 4). One possible

reason for these patterns could be increased resource availability due to decreased competition. The D+F plots in

which these three species were present experiences the highest biomass losses due to mortality during the four years

of experimental manipulation (Table S5; Fig. S11). Even though it is hard to determine the cause of death, an initial

spike in tree mortality has been observed in a long-term throughfall exclusion experiment in the Amazon (Costa et

al., 2010), which also caused an increase in growth rates of remaining trees (Rowland et al., 2015). Interestingly,

Meir et al. (2018) found that tree growth and mortality in the same experiment reached an equilibrium in the long-

term (> 10 years), reporting similar values to trees in a 1 ha plot without rainfall manipulation. The lack of

responsiveness in the F and D plots, in addition to the biomass losses in some of the D+F plots (Table S5), supports

the idea that the availability of resources could be the cause of higher RGR$_{dbh}$ in the D+F compared to the other

treatments (Fig. S14). The lowest RGR$_{dbh}$ were found in plots with the D treatment, with the strongest experimental

effect on *D. retusa*, *E. cyclocarpum*, and *S. glauca* (Fig. S8). These results are very similar to what has been found

in other tropical throughfall exclusion experiments (Meir et al., 2015), in which there is an overall negative effect in

tree diameter growth by a decrease in soil moisture.

**4.3 Belowground responses**

The fertilized plots showed no nodule production. This observed trend suggests that nutrient addition

alleviates limitations for legumes (Toro et al. 2022), and confirms the facultative nature of nodulation (Barron et al.,

2011). On the other hand, nodule production was the highest for both drought treatments (D and D+F). In part, a

decrease in soil moisture slows down the rate of nitrogen mineralization and limits plant nutrient uptake (Borken and

Matzner, 2009; He and Dijkstra, 2014). Comparable to our results, the legume species *Robinia pseudoacacia* L. also

increased nodulation in a drought experiment (Wurzburger and Miniat, 2014). Moreover, trees tend to rely more on

deeper water sources with less access to nutrients (Querejeta et al., 2021), which might also enhance nodulation in

legumes. Collectively, our data and these studies suggest that the effects of soil moisture reduction go beyond

ecosystem water/carbon balance and could cause a domino effect that might alter forest biogeochemistry.

Our soil incubation results suggest that global change has the potential to alter microbial CUE and the

susceptibility of soil carbon to pulse rainfall events in tropical dry forests. After three years of treatment, soil

microbes in the D and F soils had significant increases in CUE (Figure 4). Increases in CUE are commonly

attributed to shifts in the microbial community (Domeignoz-Horta et al., 2020) or a reduction in carbon investment

by microbes in enzymes to fuel the nutrient acquisition (Manzoni et al., 2012). In this experiment, however, the

increases in CUE in the D and F soils but not the D + F soils hinder our ability to narrow down which of these

mechanisms may be driving our results. Quantifying the shifts in microbial community composition, as well as the

identity of microbes that are active decomposers, may shed light on the mechanistic underpinning of the CUE

response observed here. Importantly, these differences in CUE across treatments also appeared to impact the

response of the soils to large, simulated rainfall events. Regardless of treatment, rewetting the soils to water holding





capacity led to a large reduction in CUE (Figure 4). While not statistically significant, there was a clear trend of
greater CUE declines in the treatment soils, particularly the D soils. This trend suggests that when large rainfall
events occur in disturbed soils; these decreases in microbial CUE could potentially lead to a stronger Birch Effect
and enhance the soil C loss (Schimel, 2018). In support, we found that rewetting the soils also led to the glucose
addition driving greater priming of soil carbon losses, a result that was particularly pronounced for the D soils (Fig.
S12). By contrast, the glucose addition in soils that were held at field soil moisture conditions led to the net
mineralization of soil C by the microbial community. Collectively, our soil incubation results highlight a critical
need for more research on the potential for global change to lead to shifts in microbial community composition and
traits in TDFs.
**5 Conclusions**

Our results highlight that forest productivity responses to is sensitive to soil fertility and that this modulates
how TDFs tree species respond to reductions in soil moisture. However, despite adding both macro- and micro-
nutrients, our results confirm that the short-term responses of tropical trees to fertilization treatments are modest at
best. At the same time, the nodulation data indicate that there might be a tight coupling between nutrient availability
and water availability in this system. Studying the role of soil moisture on plant nutrient acquisition dynamics
remains a largely unexplored venue in TDF ecology. Moreover, little is known of how these belowground processes
interact with microbial community dynamics, such as CUE, also affected by nutrient additions or reductions in soil
moisture. Beyond these processes, disentangling the causes and consequences of colimitation by water and nutrients
in productivity could help to elucidate how future climatic conditions will affect carbon cycling in the TDF.
**6 Data availability**

The data reported in this publication and associated R code for statistical analysis can be found in the
following DRYAD repository: Vargas G., German et al. (2022), Throughfall exclusion and fertilization effects on
tropical dry forest tree plantations, a large-scale experiment, Dryad, Dataset,
https://doi.org/10.5061/dryad.5x69p8d6r
**7 Author contribution**

GVG, BGW, DM, DPA and JSP designed the experiments. GVG, DPA, LDPA, DPAR, JTT and NR
performed field measurements. NR and EM performed CUE laboratory measurements. GVG processed the field
data. GVG and NR performed statistical analyses with input from EB and JSP. GVG wrote the initial draft with
input from JSP. All authors contributed with edits and feedback in subsequent versions.
**8 Competing interests**





David Medvigy is a member of the editorial board of Biogeosciences. The remaining authors have no
conflicts of interest to declare.

## 9 Acknowledgements

We thank the United States Department of Energy for funding through the research grants DE-SC0014363
and DE-SC0020344. We also thank the Explorers Club Washington Group graduate student research grant assigned
to Nanette Raczka. We thank for outstanding support in the field from Julio Zúñiga Marín, Laura Toro Gonzáles,
Erick Calderón Morales, Pedro Alvarado, Duncan Coles, Caroline Bray, Michelle Monge Velasquez, and Ronny
Hernández. We also thank for logistical support to Milena Gutiérrez-Leitón and all the staff of the Estación
Experimental Forestal Horizontes. Additionally, we thank Dr. Joseph S. Wright for constructive feedback in a
previous version of this manuscript. This research was done in accordance with Costa Rica's Ministerio Nacional de
Ambiente, Energía y Telecomunicaciones.

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

**TABLES AND FIGURE CAPTIONS**
**Table 1.** Focal tree species present in the experimental manipulations and their functional and hydraulic traits
measured in other studies (Data from Powers and Tiffin, 2010; Powers et al, 2020). Here we present species leaf
habit (LH), nitrogen fixation (NF), specific leaf area (SLA, cm$^2$ g$^{-1}$), wood density (WD, g cm$^{-3}$), water potential at
turgor loss point ($\Psi_{TLP}$, Mpa), and the water potential at 50 % accumulation of embolisms ($\Psi_{P50}$, Mpa).

| Family | Species | LH | NF | SLA | WD | $\Psi_{TLP}$ | $\Psi_{P50}$ |
|---|---|---|---|---|---|---|---|
| Bignonaceae | *Handroanthus impeteginosus* (Mart. ex DC.) Mattos | DC | N | 97.85 | 0.71 | -1.95 | -3.15 |
| Fabaceae | *Dalbergia retusa* Hemsl. | DC | Y | 67.70 | 0.80 | -1.99 | -4.71 |
| Fabaceae | *Enterolobium cyclocarpum* (Jacq.) Griseb. | DC | Y | 145.51 | 0.38 | -1.75 | -2.73 |
| Fabaceae | *Hymenaea courbaril* L. | SD | N | 69.45 | 0.84 | -1.91 | -4.2 |
| Meliaceae | *Swietenia macrophylla* King. | DC | N | 68.86 | 0.67 | -1.65 | -2.92 |
| Simaroubaceae | *Simarouba glauca* DC. | EV | N | 54.89 | 0.41 | -1.98 | -2.81 |






**Figure 1. a)** Geographical location and layout of a throughfall exclusion by fertilization experiment in Northwestern
Costa Rica. **b)** Picture of a throughfall exclusion structure in a 30-year-old *Swietenia macrophylla* King. and
*Hymenaea courbaril* L. plantation.

**Figure 2.** Volumetric soil moisture records at two depths for plots with a throughfall exclusion structure and plots
without it. **a)** Temporal variability at a weekly resolution median volumetric soil moister with its associated 75 and
25 percentiles, where the dashed vertical line represents the date when the structures were established. **b)** Histogram
distribution of the percent difference between soil moisture during the experiment ($SM_{Exp}$) and the soil moisture
preceding the experimental treatments ($SM_{Pre}$) for each depth during the dry season and the wet season, where the
vertical lines represent the median $SM_{Exp}$ - $SM_{Pre}$ percent value for plots with a throughfall exclusion structure
(dashed) and plots without it (continuous). Reported results from a linear mixed effect model comparing weekly
$SM_{Exp}$ - $SM_{Pre}$ percent values for each depth during the dry season and the wet season.

**Figure 3.** Relative growth rate responses ($RGR_{DBH}$) of plantation (panel a) and understory (panel b) trees to
fertilization (F), drought (D), and drought plus fertilization (D + F) over a period of four years (2016-2020). Bar
plots showing the media with the associated standard error (error bars) were obtained from a total of 194 plantation
trees and 462 understory trees in 16 experimental plots. Lowercase letters stand for multiple comparisons among
experimental treatments from a Post-Hoc Tukey's honest significance difference test.

**Figure 4.** Microbial carbon use efficiency (CUE) and the log-response ratio between lab rewet and non-rewet
samples ln(RR) in control, fertilization (F), drought (D), and drought plus fertilization (D + F) during the wet season
of 2019. Panel a) shows bar plots with the mean response with the associated standard error (n=4) and panel b)
shows interaction plots among experimental treatments. No significant differences were present after performing a
Post-Hoc Tukey's honest significance difference test, despite the evidence of a moderate effect of F and D in both
CUE and ln(RR).

**Figure 5.** Responses of ecosystem net primary productivity (NPP), aboveground net primary productivity (ANPP),
and aboveground to belowground ratios to fertilization (F), drought (D), and drought plus fertilization (D + F) over a
period of four years (2016-2020). Panel a) shows median values for each experimental manipulation with their
associated standard error (n=4) with significance values after performing a Post-Hoc Tukey's honest significance
difference test where $p < 0.05$ (*) and $p < 0.1$. Panel b) shows the interactions between F and D treatments where for
NPP and ANPP there was a greater response of non-drought plots to fertilization.



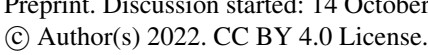


**Figure 1**



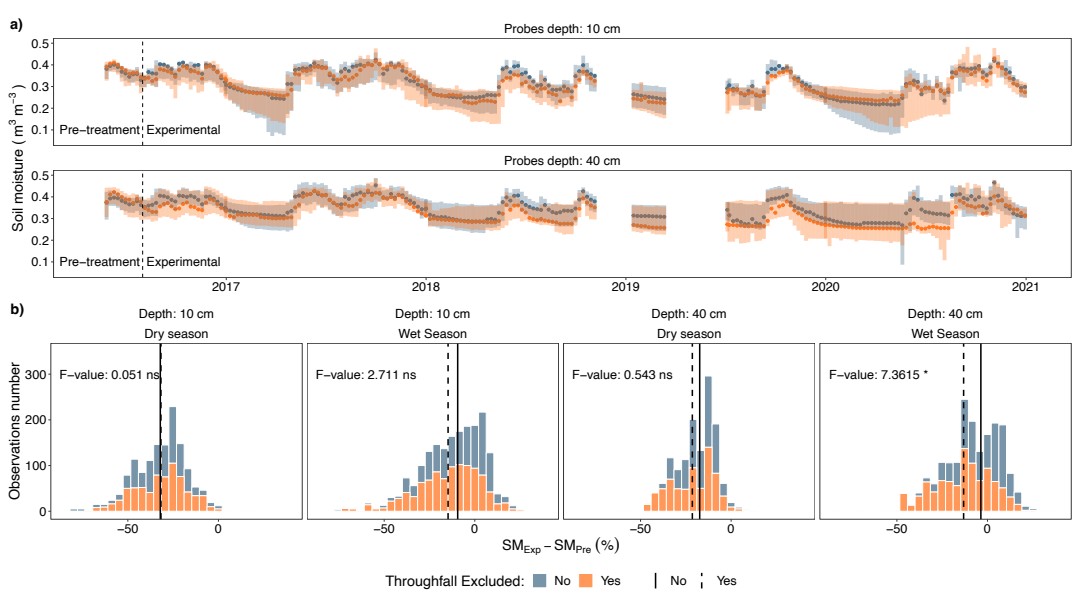


**Figure 2**








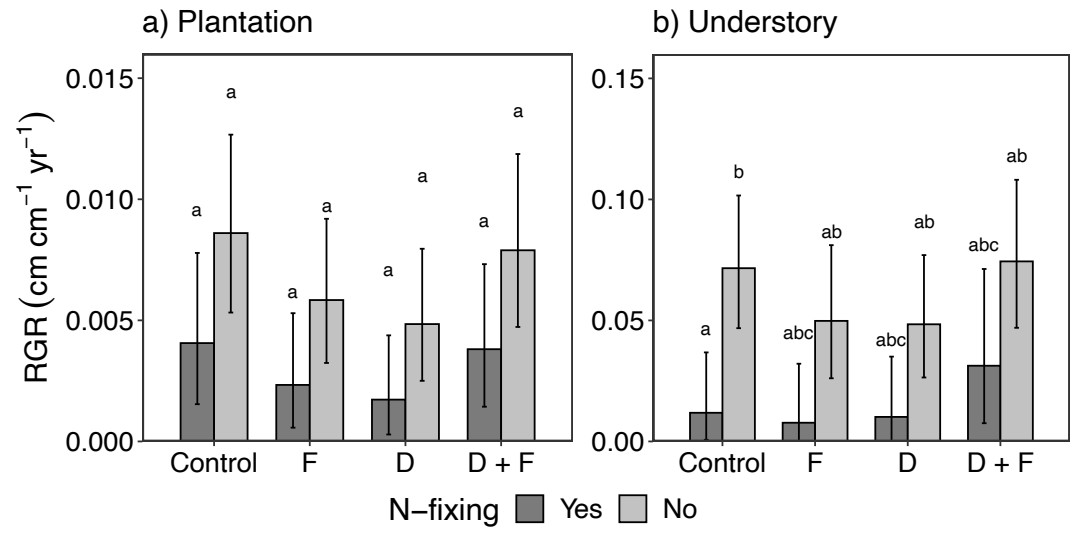


**Figure 3**



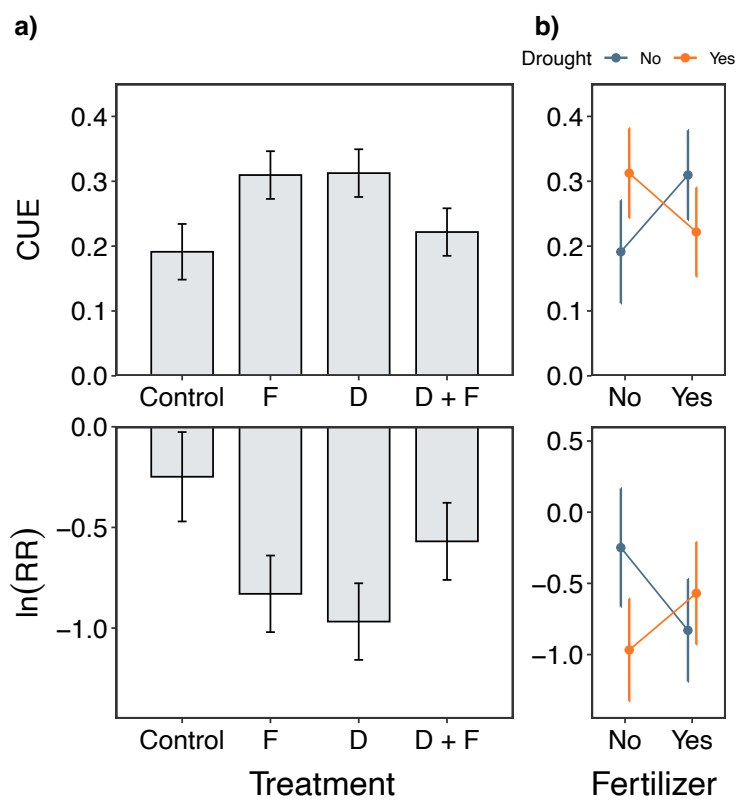


**Figure 4**





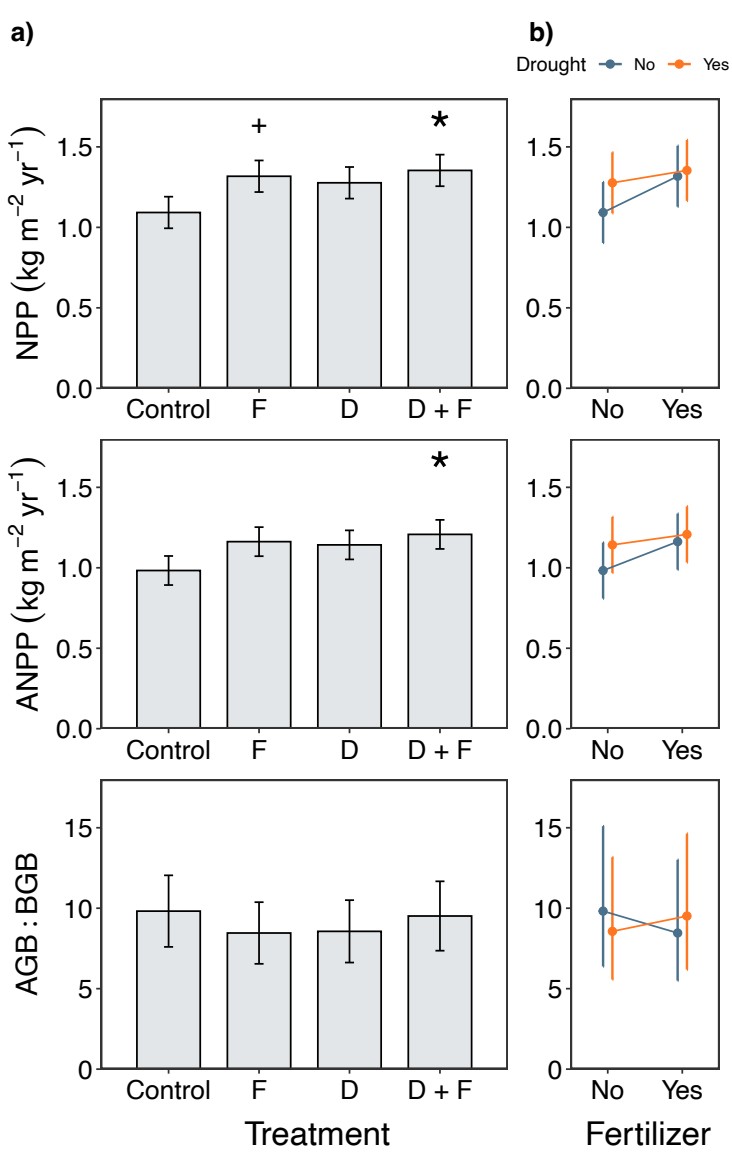


**Figure 5**