# Peer review of "Throughfall exclusion and fertilization effects on tropical dry forest tree plan- tations, a large-scale experiment"

_Biogeosciences, 2022_

## Referee Comment (RC2)

I applaud the substantial effort that went into collecting and analyzing this data. This manuscript sets out to determine how drought and nutrient-status of soil interact to control tropical dry forest carbon cycling processes. While this is a meaningful dataset with great potential, I have considerable concerns with its current presentation. The broad scope of the response variables measured, and the complexity of the study ecosystem have come together to create a rather murky message.

My primary concern is on the statistical robustness of tree-species effects. The effective unit of experimental replication for tree-species effects appears to be one. There is no description of how this is dealt with in the statistical analysis, in addition there are results reported on N-fixation and leaf-habitat effects of tree species, which are not evenly distributed across the six species they target, and also, not addressed in the statistical analysis section. An infographic is needed to breakdown what effects are tested with what level of replication. The authors may find the structure of a Before-After-Control-Impact study helpful.

Overall, the paper lacks a strong central theme, is this a manuscript about carbon cycling in tropical dry forests? Or primarily about global change impacts on tropical dry forests? Or both? Currently it reads primarily as a forest study, with some soil microbial ecology added in. It may be easier to focus the manuscript if the tree responses and the microbial responses are reported in two separate manuscripts (just some food for thought). In a shameless act of self-promotion, the authors may find this paper helpful (and no insult will be taken if not):

https://doi.org/10.1016/j.soilbio.2022.108680

Lastly, is nutrient deposition indeed an issue in most tropical dry forests? It seems that the role of innate variation in soil fertility is a more compelling justification of the study, which would be greatly bolstered by pre-treatment nutrient data. After graphing the post-treatment soil C:N data provided in the supplement, it seems that soil nutrients strongly co-vary with certain tree species. Pre-existing variation and the history of the plantation need to be more thoroughly addressed to give context to the manipulation.

[Figure]

Line comments:

67) because? rather than 'as'?

70) first use of this undefined abbreviation

Section 1.1 This is a little chicken before the eggish. Inverting the paragraph will allow you to first establish that innate variation in soil fertility drives variation in growth-response to rainfall. Then second, the opposite is also true (nutrient limitation negatively affects water use efficiency). This then logically leads to the third point that high nutrient availability could alleviate drought stress BUT at a potential cost to NPP (which is where I assume you are headed).

74) remove 'main'

80) replace the slash with 'or'

82) "Specifically", rather than "Moreover"

89) Why are transpiration rates increasing? Is this a generally accepted phenomenon in response to nutrient deposition?

94) Perhaps "current thinking" would be more appropriate than "Theory", as the later implies an established and tested precedent in the field

259) Priming is of interest here too?! Unless there is a specific hypothesis that you are testing, I don't see that this adds to the manuscript.

294) This is a major flag, if soil properties were so variable that something as basic as volumetric water content cannot be compared across plots, how are any of the other comparisons valid? Especially given that the previous sentence claims that soils were saturated when the plots were established – does this mean that the volumetric content at saturation was meaningfully different for each plot?
The current approach requires either a strong validation or for the analysis to be shown using both response variables, allowing the reader to weigh the value of each.

Table S2 as two sets of boxplots, one by treatment and one by stand type

309) Below 0, I believe is meant.

328) Love the language of evidence used here – please use this throughout!

329) Is the '-v' convention a requirement of the journal? It's a little distracting. This may make a better small table, rather than a string of text, or similar information included in the infographic mentioned above.

I'd like to conclude by saying that I really do think this dataset has great promise, and believe that honing the message will make it easier to digest and therefore more widely cited. Best of luck.

---

## Author Response (AR1)

School of Biological Sciences
South Biology Building
257 S. 1400 E., Salt Lake City, Utah 84112

7th February 2023
german.vargas@utah.edu

Dear Dr. Bahn,

We thank you for the opportunity to submit our revised manuscript bg-2022-203 *"Throughfall exclusion and fertilization effects on tropical dry forest tree plantations, a large-scale experiment"*. We also want to thank the reviewers and the Associate Editor Dr. Richard Nair for the constructive feedback provided that allowed us to streamline the manuscript and tell a holistic story on how nutrient availability and soil moisture interact to control above- and below-ground ecosystem processes. In the revised version you will notice improvements in the following areas:

- A more cohesive story on the importance of nutrient and water availability in mediating ecosystem processes that describe carbon cycling.
- A re-defined central aim of the study aimed to facilitate an easier story to follow.
- Added clarity in the description of the methods.
- Added clarity in the description of the analysis to test the effects of throughfall exclusion on soil moisture.
- Addition of suggested citations in the introduction and the discussion.

Below we detail the changes that we made to the manuscript in relation to these points. For clarity, the reviewers' suggestions appear in plain text, and our responses appear in blue text. If there is any additional way that I can facilitate the publication of our manuscript, do not hesitate to contact me. Thank you for your attention.

Sincerely,

German Vargas Gutiérrez, Ph.D.
NOAA C&GC Postdoctoral Fellow
School of Biological Sciences
The University of Utah
Salt Lake City, UT 84112

**Referee: 1**

This manuscript aims to tackle one very important question related to how climate change will affect tropical (dry) forests: the actual role and interaction between water and nutrient availability controlling forest growth. The authors studied different aspects of aboveground and belowground plant/soil components in a factorial throughfall and fertilisation experiment in a tropical dry forest in Costa Rica. This is a first for tropical forests and this fact alone already grants the manuscript a high interest for the scientific community. Besides that, I believe that the manuscript is a good fit for this journal as well. Due to the high number of studied variables (which is great, of course) and the experimental design that has the effect of drought, nutrients, tree species, N-fixing capacity and canopy versus understory trees, some parts of the text get a bit hard to follow. Nonetheless, because of such complexity, I acknowledge that the authors did a very good job in reporting the results, both in terms of the text, but also figures/tables. Supplementary material also has some important extra data that made it easier to grasp some of the reported trends (or lack of). As you will see in my comments below, I have some technical small comments on writing (very minor) and some comments on some things that are not clear as they are in the text at the moment. I hope the authors find them useful to improve the manuscript. In my opinion, one weakness is that although there are many non-significant or marginally non-significant results/trends, the authors still base a lot of the discussion and conclusions on those. I understand that there are not really a lot of strong responses, especially with drought, but I suggest then the authors tone down some statements, and for the purpose of improving future science, acknowledge and suggest where and how such an experiment could be better replicated in the future.

We want to thank the referee for the time and effort invested in the constructive review of our work. We appreciate all the suggestions and the points where clarification was needed. Indeed, in our study, we found many non-significant results. Despite that, we did find some treatment effects, even after accounting for the factor contributing the most to plot-to-plot heterogeneity: plantation stand. In the corrected version of the manuscript, we are incorporating the suggestions and making clear the scope of inference of our results (significant and non-significant) which has been highlighted by both referees as the main area of improvement. We are also updating the conclusions to include recommendations based on what we have learned from this experiment, and how to improve large-scale drought experiments.

Point-by-point comments:

- Line 21: In the abstract, I had the impression that some results that are later reported as trends but are not significant are emphasised here. Perhaps in this way, the abstract is overselling the story a bit.

We appreciate this suggestion by the referee with which we agree that the text in the abstract should be clear about the non-significant results. We modified the abstract according to the scope of inference of the observed results. This required a modification of the entire text in the abstract:

"Across tropical ecosystems, global environmental change is causing drier climatic conditions and increased nutrient deposition. Such changes represent large uncertainties due to unknown interactions between drought and nutrient availability in controlling ecosystem net primary productivity (NPP). Using a large-scale manipulative experiment, we studied for four years whether nutrient availability affects the individual and integrated responses of above- and below-ground ecosystem processes to through-fall exclusion in 30-year-old mixed plantations of tropical dry forest tree species in Guanacaste, Costa Rica. We used a factorial design with four treatments: control (CN), fertilization (F), drought (D), and drought+fertilization (D+F). While we found that a 13-15% reduction in soil moisture only led to weak effects in the studied ecosystem processes, NPP increased as a function of F and D+F. The relative contribution of each biomass flux to NPP varied depending on the treatment, with woody biomass being more important for F and root biomass for D+F and D. Moreover, the F treatment showed modest increases in maximum canopy cover. Plant functional type (*i.e.*, N-fixation or deciduousness) and not the experimental manipulations was the main source of variation in tree growth. Belowground processes also responded to experimental treatments, as

we found a decrease in nodulation for F plots and an increase in microbial carbon use efficiency for F and D plots. Our results emphasize that nutrient availability, more so than modest reductions in soil moisture, limits ecosystem processes in tropical dry forests and that soil fertility interactions with other aspects of drought intensity (*e.g.*, vapor pressure deficit) are yet to be explored."

- Line 112-113: This reference could come in the introduction but also it could be relevant for some parts of the discussion of the results.
  https://www.nature.com/articles/s41586-022-05085-2

We agree with the referee on the pertinence of the article by Cunha et al. (2022) to our study and in general the importance for the field of tropical ecosystem ecology. We have now cited the article multiple times:

Lines 53-55: "Tropical forests exist in a highly heterogeneous mosaic of soil fertility, parent material, and topography (Townsend et al., 2008; Augusto et al., 2017; Waring et al., 2021), properties that shape forest composition (Condit et al., 2013; Werden et al., 2018a), and function (Clark and Clark, 2000; Cunha et al., 2022)."

Lines 75-77: "While it is well documented that tropical leaf phenological cycles depend on plant water status and the start of the rainy season (Frankie et al., 1974; Borchert, 1994), phosphorus fertilization seems to reduce leaf life span in Eastern Amazon forests (Cunha et al., 2022)."

Lines 84-86: "The increase of specific nutrients (*i.e.*, via nitrogen deposition) might cause an imbalance in stoichiometry or increase water demand, which plants will adjust by increasing transpiration rates or producing more root biomass (Lu et al., 2018; Waring et al., 2019; Cunha et al., 2022)."

Lines 108-110: "While nutrient addition experiments have shown mixed (strong, weak and none) effects on tree growth in tropical forests (Wright et al., 2018; Hou et al., 2020; Cunha et al., 2022),…"

Lines 421-422: "Such responses agree with known evidence of nutrient limitation on productivity in tropical forests (Alvarez-Clare et al., 2013; Wright et al., 2018; Waring et al., 2019; Cunha et al., 2022),…"

Lines 425-427: "Increases in root and woody productivity in response to nutrient additions have been observed in secondary wet tropical forests (Wright et al., 2018) and Eastern Amazon forests (Cunha et al., 2022)."

Lines 443-445: "However, the opposite was observed in a fertilization experiment in the same region where nutrient additions reduced leaf life span and had no effects on $LAI_{max}$ (Cunha et al., 2022)."

Lines 498-500: ". However, despite adding both macro- and micro-nutrients, our results confirm that the short-term responses of tropical dry forest trees to fertilization treatments are modest at best, contrary to the observed strong responses in nutrient-depleted eastern Amazon forests (Cunha et al., 2022)."

- Table S1: It's not clear what's the unit of the fertiliser values.

We apologize for the lack of clarity in the information presented. The values are in kg, which we have now incorporated into the table.

- Table S1 and S2: It's a minor detail but treatment abbreviation differs in the supplementary material and the main document.

We apologize for the lack of clarity in the abbreviation of the treatments. Throughout the text and the supplements, we standardized the abbreviations as Control, Drought (D), Drought+Fertilization (D+F), and Fertilization (F).

- Figure S4: It seems that the significant difference in LAI in 2017 was found in all treatments, but only DR and FR are discussed in figure caption. Can you clarify here, please?

This figure shows the effect of Hurricane Otto during the month of November in the LAI data. We apologize for the lack of clarity in the text and now made clear that the effects were present across treatments. The figure caption reads:

"**Fig. S4.** Leaf area index mean values with 95% confidence intervals for each experimental treatment (Control, D+F: Drought+Fertilizer, D: Drought, F: Fertilizer) during the month of November. We found a significant year effect (F=21.29; d.f.= 3; p < 0.001) because of tropical storm Nate, in which LAI decreased during November 2017 across all treatments. Letters represent multiple comparisons among years within experimental treatments from a Post-Hoc Tukey's honest significance difference test."

- Line 155/ Figure S2: Indicate somewhere how distant the plots are from each other.

We now indicate in the methods section the minimum and maximum distances between plots. This varied depending on the treatment, as a through-fall exclusion treatment would not cause a major effect on a plot nearby (minimum distance: 15 m apart). However, we were aware that fertilizer can leach away to other plots. For this reason, we took two measures to avoid this: 1) we placed fertilized plots more than 50 m away from other plots, and 2) we always placed fertilizer plots down the slope of the nearest plot. Despite the terrain being relatively flat, there is a small slope that facilitates such an approach. These measures were corroborated when observing water flow during big rainfall events. We apologize for the lack of clarity in this regard. The text in lines 169-172 now reads:

"We placed fertilized plots more than 50 m away from other plots or down the slope from control and drought plots whenever we could not find enough trees 50 m away. These measures were considering the possibility of nutrient leaching from one plot to another one."

- Line 159: Any specific design for soil sampling (e.g. corners/centre of the plot)?

We now incorporated the following text in lines 151-154:

"Soil samples (0-10 cm depth) were collected in 2016 and 2021, by taking 7 to 10 cores (2.5 cm diameter, one on each corner and three to six in the center line of the plot) and compositing cores by the plot. Particle size distribution was collected in 2016 (Table S2), extractable elements (Olson extractable Cu, Zn, Mn, Fe, K, and P), and total C and N for samples collected in the fifth year (2021)."

- Line 175: Is there any indication that the study sites were more N or P limited? I see the focus of having comparable N concentrations (like the experiment in Panama), but I just wonder about the other elements, why the choice of a broad-spectrum fertiliser?

In general, the soils in the area tend to be P limited as highlighted by Waring et al. (2019), and shown in Table S2 and Fig S7. Regarding the fertilizer choice, there are a couple of factors that influenced our decision. First, as with other fertilization experiments (Wright et al., 2011; Alvarez-Clare et al., 2013; Waring et al., 2019) responses can be limited by multiple factors. Second, in nearby secondary forests where we studied how increases in rainfall lead to increases in net primary productivity (ANPP), we have found that soils with lower total elements (P, N, and cations except Mn), lower cation exchange capacity, and lower pH tend to limit how ANPP increases with higher precipitation (Becknell et al., 2021). This suggests that multiple nutrients might be limiting such responses. Since our interest was to understand how nutrient availability limits responses to soil moisture, we decided to use a broad-spectrum fertilizer rather than looking at the interaction of different nutrients with soil moisture changes. Our goal was to

boost nutrient availability overall, and hopefully overcome the limitation of any nutrient, irrespective of the identity of that nutrient. However, we acknowledge that this limits the potential of pointing out which specific nutrient limit plants' responses to soil moisture.

- Line 185: I thought that due to its tree-centric approach in building the plots, only the planted trees were going to be taken into account when measuring productivity. Couldn't the panels at the throughfall exclusion limit new plant appearance/recruitment?

Panels can limit new plant appearance recruitment indeed. However, we placed the panels in a way that we avoided the standing understory plants. Regarding recruiting understory plants, we did move some of the panels in two plots to allow plants to grow. In this case, it only required moving one panel 15-30 cm to the right or the left. Whenever we did that, we placed a portion of a panel on the other side of the recruiting tree. However, it is possible that in the future this can be evaluated, as it is a commonality and a limitation among throughfall exclusion experiments.

- Line 200-201: There is, perhaps, a mistake in this sentence, as "leaves, and, reproductive litterfall" appear twice in the description.

We corrected the text to add clarity. Now the text on lines 194-196:

"We then calculated the annual litterfall productivity in kg m$^{-2}$ yr$^{-1}$ for total litterfall (leaves, small branches, flowers, and fruits), only leaves, and reproductive litterfall (flowers and fruits)."

- Line 238: I understand how difficult it must be to sample these cores in the dry season. One thing that needs clarification here then, is if root productivity in kg m-2 yr-1 was calculated extrapolating those 2-month interval sampling periods to one year, or if root productivity between, for example, November to June next year was taken into account. If the second is true, I would imagine that during these 6 months root productivity might not be accurate, as root mortality and recruitment could have happened. On the other hand, by using root productivity only during the wet/growth season, perhaps root productivity on a yearly basis could be overestimated (assuming there is higher root productivity in the wet and lower in the dry season). In year 1, you sampled along the whole year, right? How does the data between different seasons compare for this year?
- Another point, so in every sampling, you installed new ingrowth cores in different locations. Did I get that right? If so, would you also have root stocks in those same sampling dates, or when freeing the soil from roots to install the ingrowth cores, these roots were discarded? Just curious to know a bit more about the root dynamics here.

We appreciate the interest of the referee in our root sampling protocol. In these highly seasonal forests, both belowground and aboveground biomass is limited by rainfall, and during the dry season the production of roots is minimal or non-existent (Kummerow et al., 1990; Kavanagh and Kellman, 1992; Waring et al., 2019). However, to avoid overestimating or providing inaccurate data on root productivity as highlighted by the referee, we installed the root ingrowth core that was going to be collected in June during the November collection. In this way, we did consider root growth, if any, during the dry season. The one limitation is that we cannot make inferences on root seasonal dynamics. However, that was not the objective of the experiment as we focused only on the total annual flux. We installed the new ingrowth cores in the same location within the plot (corners and center line), but in a different soil core. We indeed discarded all plant/fungal material from the soil core when preparing the new ingrowth core to ensure we measured new root growth. However, we did not quantify the weight of roots present in those soil cores when preparing the new ingrowth core.

To make these points clear we modified the text in lines 228-239 as follows:

"The cores were collected two months after deployment and a subsequent new set of cores was installed right after collection. While deploying the cores, we filled them with sieved, root-free soil collected on-site. During the first year of the experiment, cores were sampled in the dry season. However, the clay-rich soils harden greatly during the dry season, which increased the difficulty of deploying new bags during these times. For the following three years, ingrowth bags were harvested in June, August, and November, with the modification that the bags harvested in June were deployed in November. We acknowledge that roots may have grown, died, or decomposed during the dry season (Kummerow et al., 1990). However, this effect will lead to minimal bias in annual productivity totals, as dry season root growth and decomposition are expected to be negligible in the TDF (Kavanagh and Kellman, 1992). After collecting the cores, fine roots were separated from the soil by washing them over a 2 mm sieve. We counted the number of nodules on each root sample if present. Finally, root samples were dried for 72 hours at 60 °C and weighed to estimate total fine root productivity in kg m$^{-2}$ yr$^{-1}$."

- Line 315-316: Could you please add to this sentence if such reported differences are statistically significant?

We now modified the text to read as follows:

"At 40 cm depth, we found evidence ($p < 0.05$) of a ~ 13% reduction in soil moisture as a function of the pretreatment period in the plots with a throughfall exclusion structure, contrary to a weak ~ 4% reduction in soil moisture in the plots without throughfall exclusion (Fig. 2)."

- Lines 315-318: My main concern is related to the extent to which the throughfall exclusion really worked, and how this may have affected the general lack or weak trends you find in your study. This should be acknowledged a bit more in the discussion, perhaps in a way to provide advice for future research. Many of the results you show in the supplementary material, for example, regarding the production of flowers and seeds, are marginally non- significant, despite the big difference in productivity. I imagine that in addition to specific species differences, the relative short nature of the experiment was not enough to capture significant changes.

Our response to this comment is twofold. First, we agree here with the perspective of the referee and we tried to touch on this point from the perspective that perhaps the observed changes in soil moisture are not strong enough to induce significant changes in productivity patterns in response to soil moisture. Second, we did not manipulate atmospheric water demand (e.g., VPD) and this may be an equally important component of drought than soil moisture. Throughout the discussion, we touched on this issue multiple times now. We also added a conclusion with recommendations on how to impose greater drought stress:

Lines 393-394: "We found that a 13-15% reduction in soil moisture only leads to modest effects in the studied ecosystem processes."

Lines 400-402: "Collectively, these results suggest that reducing soil moisture by a modest amount is not sufficient to drive large reorganizations in ecosystem processes, and that soil nutrient availability mildly modulate short-term changes in productivity."

Lines 405-419: "In a broad sense, we found that nutrient availability had a stronger control on forest productivity than a ~13-15% reduction in soil moisture. While this result does not resonate with the expectation that water availability imposes a greater limitation on productivity across environmental gradients than soil fertility (Harrington et al., 1995; Santiago and Mulkey, 2005; Toledo et al., 2011; Sala et al., 2012; Poorter et al., 2016), it provokes the question to what extent are tropical dry forests resilient to drought stress? Our data point to other aspects related to drought intensity and not soil moisture alone that could be key factors in how water availability shapes TDF primary productivity (Anderegg et al., 2013). Recent studies from northwestern Costa Rica have shown that abnormal drought stress due to a strong ENSO event in 2015 caused biomass loss due to an increase in tree mortality, a decrease in reproductive biomass production, and reductions in productivity (O'Brien et al., 2018;

Castro et al., 2018; Powers et al., 2020). The main characteristics of the 2015 ENSO were the elevated temperatures and a substantial rainfall reduction for the region (Santoso et al., 2017), which can increase the severity of drought effects in forest ecosystems due to increased atmospheric water demand (Brodribb et al., 2020; McDowell et al., 2020). Thus, while throughfall exclusion experiments manipulate soil moisture, it is possible that a combination of factors such as the vapor pressure deficit, the rainfall patterns (intensity and seasonality), and their linkages to soil moisture, is a more important aspect of drought stress for forested ecosystems."

Lines 501-504: "Studying the role of soil moisture on plant nutrient acquisition dynamics remains a largely unexplored venue in TDF ecology. Considering the observed patterns, a total throughfall exclusion will be necessary to cause soil moisture to decrease by greater than 15 % and manipulations of the atmospheric water demand (*e.g.*, vapor pressure deficit) could help to improve our understanding of drought in tropical forests"

- Line 321: Since there were not such big differences between mean soil moisture comparing control plots and throughfall exclusion, I wonder if this signal is small because of a potential effect of the years, meaning that perhaps stronger patterns could be seen towards the end versus the start of the experiment. Have you tested for the changes in soil moisture between years and treatments?

Besides the control plots (Fig. S6), we did not evaluate the year-to-year soil moisture variation in all the treatments. However, in the analysis of throughfall exclusion effects on soil moisture, we included the week of the year for a given year as part of the analysis as we acknowledge there is intra-annual variation in soil moisture associated with the seasonality of the tropical dry forest (Schwartz et al., 2022). It is important to notice that we did this separately during the wet and dry seasons. We are making sure that this is clear in the revised version.

In the notation for the R package 'lmer' the model was written as follows:

lmer(formula = sm_perc ~ TFES+time + (1|plot)+(1|time:plot),
                REML = TRUE,
                data= sm_long2.dry10,
                control = lmerControl(optimizer ='optimx', optCtrl=list(method='nlminb')))

Where
sm_perc is the % change in water content
TFES is the presence of throughfall exclusion (yes or no)
time is the week number of a given year for either the wet or the dry season, in the above example the dry season.

- Line 330-331: Can you please add to this sentence if such interaction resulted in a positive or negative trend?

The interaction occurred in that when applying fertilizer trees in drought plots showed an increase in RGR while trees in non-drought plots a decrease, particularly for non-N-fixing plants. The text now reads:

Lines 330-333: "We found moderate evidence of an interaction between drought and fertilizer for plantation trees ($F$ = 5.16, *d.f.* = 1, *p* = 0.0499) and weak evidence for understory trees ($F$ = 5.04, *d.f.* = 1, *p* = 0.0659) (Fig. 3), whereas the effects of fertilization caused an increase in RGR for trees in drought plots and a decrease for trees in non-drought plots (Fig. S8)."

[Figure]

Fig. S8. Relative growth rates (RGR) for plantation trees. Mean plots with their associated standard error show the interaction ($F = 5.16$, *d.f.* $= 1$, **$p = 0.0499$**) between the effects of fertilizer and drought treatments on nitrogen-fixing (N-fixing) and non-nitrogen-fixing (Non-N-fixing) plant functional types.

- Line 870: I think it should read "median", not "media" (or mean?). Same for some figures in the supplementary material.

We fixed this mistake in all the figures as it should read the mean (in the case of Figure 3).

- Line 356: There is a discrepancy between the unit described for root productivity in the methods section (annual basis) and here in the results.

We added and corrected the units as mistakenly we omitted the year. It now reads (kg m$^{-2}$ yr$^{-1}$)

- Line 381: You state there are no significant differences but the p value is 0.0431, can you clarify this please?

Indeed, the ANOVA test used reported a p-value < 0.05. However, when studying the pairwise comparisons with Tukey's HSD we found no differences between the drought and the control treatments. Now the text reads:

"Although we observed a 14% NPP increase in the drought plots ($F = 5.29$, *d.f.* $= 1$, **$p = 0.0431$**), we found no evidence this was different from the control plots after looking at the multiple comparisons (Fig. 5)."

- Lines 421-423: Great argumentation here!

Thank you!

- Line 426: Again, I think that this recent paper could make it to the discussion (strong and direct evidence of P limitation in Amazon forests). This paper contradicts the statement you also make on line 500 in the conclusion, since the authors found strong NPP responses after 2 years of fertilisation in a mature forest. https://www.nature.com/articles/s41586-022-05085-2

Please refer to answer to comment on lines 112-113.

- Lines 434-436: I would suggest toning down this sentence a bit, as this is a trend, and no real strong evidence of colimitation by water and nutrients were found in your study.

We understand the point made here by the referee and apologize for the lack of clarity, given the fact that we did not find a strong interaction signal. We modified the text, it now reads:

"Moreover, the increase in productivity as a function of fertilization showed a bigger, yet not significant, increase without the presence of throughfall structures (Fig. 5, panel b). This trend resembles observed patterns in nearby stands of TDF, where forests in fertile soils are more responsive to increases in rainfall than forests in infertile soils (Becknell et al., 2021)."

- Line 441: Insert . after "treatments".

Thank you for pointing out this mistake. We modified the text accordingly.

- Line 456: Better to use the past tense here: "experienced".

We fixed the grammatical mistake.

- Line 463: Could you specify which resources you refer to here, perhaps light?

Although we cannot point out a single resource, light availability is one of the possible resources limiting plant growth. We modified the text to state that in those plots there is in general less competition. The text in line 460 now reads:

"...that the availability of resources such as light could be the cause of higher $RGR_{dbh}$ in the D+F…"

- Line 475: It could be useful to acknowledge the fact that in these dry forests, and especially by increasing drought experimentally, roots can go really deep/deeper in search for water and nutrients. The lack of responses found for root productivity in your study is limited to the 0-15cm, and if we think that you only captured changes in soil moisture at the 40cm depth, it's plausible that roots could be changing down the soil profile as well.

This is a really important point highlighted by the referee, to which we agree. Quantification of the vertical root profile in relation to experimental treatments is certainly one of the next steps in our future work in this experiment. We modified the text in lines 472-475 now reads:

"Moreover, trees tend to rely more on deeper water sources with less access to nutrients (Querejeta et al., 2021). This allocation of root biomass might also enhance nodulation in legumes as there might be changes in the vertical profile of nutrients in the soil. However, the lack of data on root production beyond the top 15 cm in our experiment makes it hard to confirm this is the case."

- Line 490: Maybe replace ; by , after "disturbed soil".

We fixed the grammatical mistake.

- Line 498: It seems there is some word missing between "responses to" and "is sensitive to".

We modified the text to add clarity and a better flow of ideas:

"Our results highlight that forest productivity is sensitive to soil fertility and that this might interact with changes in soil moisture."

**Referee: 2**

I applaud the substantial effort that went into collecting and analyzing this data. This manuscript sets out to determine how drought and nutrient-status of soil interact to control tropical dry forest carbon cycling processes. While this is a meaningful dataset with great potential, I have considerable concerns with its current presentation. The broad scope of the response variables measured, and the complexity of the study ecosystem have come together to create a rather murky message.

My primary concern is on the statistical robustness of tree-species effects. The effective unit of experimental replication for tree-species effects appears to be one. There is no description of how this is dealt with in the statistical analysis, in addition there are results reported on N-fixation and leaf-habitat effects of tree species, which are not evenly distributed across the six species they target, and also, not addressed in the statistical analysis section. An infographic is needed to breakdown what effects are tested with what level of replication. The authors may find the structure of a Before-After-Control- Impact study helpful.

Overall, the paper lacks a strong central theme, is this a manuscript about carbon cycling in tropical dry forests? Or primarily about global change impacts on tropical dry forests? Or both? Currently it reads primarily as a forest study, with some soil microbial ecology added in. It may be easier to focus the manuscript if the tree responses and the microbial responses are reported in two separate manuscripts (just some food for thought). In a shameless act of self-promotion, the authors may find this paper helpful (and no insult will be taken if not):

https://doi.org/10.1016/j.soilbio.2022.108680

Lastly, is nutrient deposition indeed an issue in most tropical dry forests? It seems that the role of innate variation in soil fertility is a more compelling justification of the study, which would be greatly bolstered by pre-treatment nutrient data. After graphing the post-treatment soil C:N data provided in the supplement, it seems that soil nutrients strongly co-vary with certain tree species. Pre-existing variation and the history of the plantation need to be more thoroughly addressed to give context to the manipulation.

We really appreciate the time and effort of the referee in performing this review, which allowed us to improve our manuscript. One aspect that both referees highlighted was that the scope of inference in the conclusions we made had to be adjusted to our results. We now addressed this in the text, as we agree that the message must be clear on what the data is telling us.

We also want to apologize for the lack of clarity in the description of the statistical analysis. In our analyses, we accounted for all the known sources of variation either through fixed or random effects. We analyzed the fixed effects by evaluating the main effect of drought, the main effect of fertilization, and the interaction in a two-way factorial experimental design. We then included random effects in this model to account for other sources of variation such as plot, stand, and species identity (when analyzing tree relative growth rates). This approach ensures that we do not ignore the influence of these factors in the observed patterns. When studying the changes in RGR, we accounted for the unbalanced design and the interactions by calculating the type III sum of squares when obtaining the F values in the analysis of variance. Having said this, we agree that the approach of before-after-control impact could be helpful. However, besides soil moisture and initial soil characteristics collected one month after the panels we installed, we did not have the ability to collect pre-treatment data on productivity, although we agree that this would have been desirable. Our expectations were that treatment effects would increase in magnitude over time. We now clarify that in the revised manuscript.

We thank the reviewer for the comment on the focus of the manuscript and we have tried to present a coherent, streamlined central theme in the revised manuscript. Our focus is on the responses of TDF to a

reduction in throughfall, and whether this depends upon soil nutrient availability. During the writing of this manuscript, we considered the option of two separate manuscripts or one manuscript telling a holistic "microbial to the ecosystem" level story. We decided to put together an ecosystem-level study that quantifies responses at various biological levels, with the challenge of increased complexity on how to tell a cohesive story. We appreciate the suggested work by the referee as it has been informative to the observed patterns in our experiment.

We appreciate the work of the referee to make the point of how soil nutrients covary with certain tree species. While there is evidence of nutrient deposition in tropical forests (Wang et al., 2017; Lu et al., 2018), forest composition might also influence soil properties and vice-versa as highlighted by the referee. We were aware of this and addressed the issue in the statistical analysis by adding the plantation species combination stand as a random effect.

Point-by-point comments:

- Line 67: because? rather than 'as'?

We modified the text accordingly:

"High nutrient availability alleviates drought stress because plants with higher leaf nitrogen maximize water"

- Line 70: first use of this undefined abbreviation

After modifications to the introduction we now define the abbreviation in line 65.

- Section 1.1 This is a little chicken before the eggish. Inverting the paragraph will allow you to first establish that innate variation in soil fertility drives variation in growth-response to rainfall. Then second, the opposite is also true (nutrient limitation negatively affects water use efficiency). This then logically leads to the third point that high nutrient availability could alleviate drought stress BUT at a potential cost to NPP (which is where I assume you are headed).

We appreciate the suggestion made by the referee as this improves the flow of ideas. The text now reads:

"Soil fertility is an important factor modulating the responses of forest productivity to rainfall variation. For instance, tropical dry forest (TDF) stands growing in more fertile soils tend to show higher increases in productivity with higher rainfall than stands in nutrient-poor soils (Medvigy et al., 2019; Becknell et al., 2021). High nutrient availability alleviates drought stress because plants with higher leaf nitrogen maximize water use efficiency at the cost of photosynthetic nitrogen use efficiency (Lambers et al., 2008), which will enhance photosynthetic capacity in favorable conditions. At the same time, nutrient limitation negatively affects water use efficiency in crop species and tropical seedlings with potential costs to productivity via reductions in carbon assimilation (Santiago, 2015). Other processes besides primary…"

- Line 74: remove 'main'

We addressed the suggestion now the text in line 72 reads:

"Leaves, and more precisely canopy cover, are the center for carbon assimilation in forest ecosystems"

- Line 80: replace the slash with 'or'

We addressed the suggestion now the text in lines 77-78 reads:

"…while changes in the timing of leaf flushing or shedding may create a cascade of effects with unknown consequences…"

- Line 82: "Specifically", rather than "Moreover"

We appreciate the suggestion. However, the entire paragraph was modified to add clarity. Now the text between lines 70-81 reads:

"Other processes besides primary productivity provide insight into ecosystem responses to global environmental change. Leaves, and more precisely canopy cover, are the center for carbon assimilation in forest ecosystems. Recent evidence suggests that the patterns of leaf flushing and leaf shedding are changing at a global scale because of climate change (Piao et al., 2019). While it is well documented that tropical leaf phenological cycles depend on plant water status and the start of the rainy season (Frankie et al., 1974; Borchert, 1994), phosphorus fertilization seems to reduce leaf life span in Eastern Amazon forests (Cunha et al., 2022). A decrease in leaf canopy cover affects productivity by decreasing the photosynthetic area (Doughty and Goulden, 2008), while changes in the timing of leaf flushing or shedding may create a cascade of effects with unknown consequences, which will affect organisms that depend on these processes (Coley, 1998). Thus, quantifying the combined effects of rainfall reductions and soil fertility on leaf production is key to disentangling the interactions between primary productivity, canopy processes, nutrient availability, and climate."

- Line 89: Why are transpiration rates increasing? Is this a generally accepted phenomenon in response to nutrient deposition?

In general, with more soil fertility there is an increase in photosynthesis, hence an increase in transpiration rates, which ultimately leads to a decrease in water use efficiency (Lambers et al., 2008). Indeed, there is evidence suggesting that this is occurring in tropical forests as suggested by Lu et al. (2018) in their work on long-term responses to high nitrogen deposition in the Dinghushan Biosphere Reserve in southern China. Among their main results are:
- Acceleration of soil acidification
- Reduction of biologically available cations
- Increases in plant transpiration

- Line 94: Perhaps "current thinking" would be more appropriate than "Theory", as the later implies an established and tested precedent in the field

We appreciate the suggestion and agree with it. We modified the text accordingly. Now the text in lines 88-90 reads:

"…changes in water and nutrient availability. Current knowledge suggests that microbes with high CUE produce more biomass that upon death becomes protected from future microbial attack by adhering…"

- Line 259: Priming is of interest here too?! Unless there is a specific hypothesis that you are testing, I don't see that this adds to the manuscript.

We appreciate this comment by the referee because it allowed us to improve the introduction and the presentation of our research objectives in this study. In our study, we were looking at responses to experimental manipulation from microbes to ecosystem productivity. Besides microbial carbon use efficiency, there are other aspects of soil microbial communities affected by drought or modulated by soil nutrient availability (Ahmed et al., 2018). We now added the following text to lines 94-100:

"Other aspects of soil microbial processes may be affected by drought or modulated by soil nutrient availability (Ahmed et al., 2018). Soil priming refers to the decomposition of older recalcitrant organic matter following the soil microbial community's stimulation by adding labile organic matter (Liu et al., 2020). If drought alters patterns of fine root growth and rhizodeposition (Preece and Peñuelas, 2016), this may lead to altered priming with altered consequences of soil organic carbon storage. Identifying the extent to which shifts in nutrient and precipitation regimes alter soil carbon cycling in TDFs is critical to increasing our understanding of climate change consequences in this important biome (Knorr et al., 2005; Chadwick et al., 2016)."

- Line 294: This is a major flag, if soil properties were so variable that something as basic as volumetric water content cannot be compared across plots, how are any of the other comparisons valid? Especially given that the previous sentence claims that soils were saturated when the plots were established – does this mean that the volumetric content at saturation was meaningfully different for each plot?
- The current approach requires either a strong validation or for the analysis to be shown using both response variables, allowing the reader to weigh the value of each.

Our response to this concern is twofold.

First, we may have over-emphasized heterogeneity. In fact, out of 16 plots, there are seven clay loams, four silty clay loams, four loams, and one silty loam. Now we are including a supplementary figure with the soil texture triangle and the placement of each plot among soil types (Fig. S4). We apologize for the lack of clarity as the message did not go the way it was intended.

Second, we did include all plots in the same linear mixed effects model, but instead of comparing the volumetric water content raw values we were comparing the change in water content in reference to the wet-season and dry-season values. We decided to implement this approach as changes in the volumetric content of water provide a measure of the effect on soil moisture after the onset of the throughfall exclusion structure, an approach that has been tested in other large-scale manipulation experiments (Reid et al., 2015). Also, it is important to mention that we did include the time point in the model, which in this case was the week during the wet season of a given year. In the notation for the R package 'lmer' the model was written as follows:

lmer(formula = sm_perc ~ TFES+time + (1|plot)+(1|time:plot),
            REML = TRUE,
            data= sm_long2.dry10,
            control = lmerControl(optimizer ='optimx', optCtrl=list(method='nlminb')))

Where
sm_perc is the % change in water content
TFES is the presence of throughfall exclusion (yes or no)
time is the week number of a given year for either the wet or the dry season, in the above example the dry season.

 Once again, we apologize for the lack of clarity in the text.

The text between lines 282-294 reads:

"To test whether the throughfall exclusion structures affected soil moisture, we performed a linear mixed model with the change in soil moisture for a given plot as the response variable, the presence of the throughfall exclusion structure and the weekly timepoints from January 2017 to December 2020 were fixed effects, and probe nested within plot nested within stand as a random intercept. This approach allowed us to test the effect on soil moisture after the onset of the throughfall exclusion structures (Reid et al., 2015). We ran separate models for each depth (10

and 40 cm), and for the wet season and dry season due to the strong rainfall seasonality. To obtain the change in soil moisture per plot, we divided the observation time into two periods, a pre-treatment (May 2016 to late August 2016) that consisted of wet season soil moisture data before the shelters were set up, and an experimental period (January 2017 to December 2020). We excluded from this analysis the data collected between September and December 2016 as we finished establishing the rainout shelters three months into the rainy season. After removing outliers using the interquartile method, we calculated the median pre-treatment soil moisture ($SM_{PT}$, $m^3$ $m^{-3}$) for each probe in each plot. We then calculated the treatment effect as the percentage change between each soil moisture observation ($SM_i$, $m^3$ $m^{-3}$) and the $SM_{PT}$."

[Figure]

Fig S4. Soil particle size distribution for each experimental treatment in this study. Classification triangle according to the United States Department of Agriculture: clay (C), silty clay (SIC), sandy clay (SC), clay loam (CL), silty clay loam (SICL), sandy clay loam (SCL), loam (L), silty loam (SIL), sandy loam (SL), silty (SI), loamy sand (LS) and sand (S).

- Table S2: as two sets of boxplots, one by treatment and one by stand type

Such a figure exists for extractable elements with the treatment (Fig S8). Now, we have added the particle size distribution triangle, and the information associated with the plantation stand in Table S2.

- Line 309: Below 0, I believe is meant.

We corrected the mistake in line 309

- Line 328: Love the language of evidence used here – please use this throughout!

Thank you! Paraphrasing the work by Muff *et al.* (2022) there is much to be said rather than a dichotomous significant vs. non-significant. The referee might appreciate the mentioned work, please check the reference list at the end of this letter.

- Line 329 Is the '-v' convention a requirement of the journal? It's a little distracting. This may make a better small table, rather than a string of text, or similar information included in the infographic mentioned above.

We modified the text and deleted the "-v" from the results section. It was intended to differentiate results from a Fischer test from the fertilization (F) abbreviation, we apologize for the confusion.

- I'd like to conclude by saying that I really do think this dataset has great promise, and believe that honing the message will make it easier to digest and therefore more widely cited. Best of luck.

We appreciate all the feedback provided as it has showed us ways to improving our manuscript.

\*\*\*\*\*\*\*\*\*\*\*\*\*\*\*\*\*\*\*\*\*\*\*\*\*\*\*\*\*\*\*\*\*\*\*\*\*\*\*\*\*\*

**Editor's comments to the author(s):**

Dear Authors,

Thank you for submitting your interesting paper and your response to the referee comments. I agree with the referees that the manuscript/dataset is of considerable interest and could be suitable for publication in Biogeosciences upon revision.

However, I also concur with the reviewers that the submitted manuscript was quite unfocused and the preprint text overemphasises some of the results. The manuscript takes the ambitious route of the holistic whole system view but needs some work to focus and address weaknesses in the analyses and interpretation. This is pointed out in the reviews. Your plan to address these includes text additions, rewrites, additional analyses, and moving material from the supplementary material to the main paper. I hope this can be achieved in the revision.
Please also attend to the minor issues as you have described.

Thank you for your submission to the special issue.

**Response to the editor:** We highly appreciate the comments by the Associate Editor, as they allowed us to focus on the areas that needed improvement in our text.

**Cited bibliography in the response letter:**
Ahmed, M. A., Sanaullah, M., Blagodatskaya, E., Mason-Jones, K., Jawad, H., Kuzyakov, Y., and Dippold, M. A.: Soil microorganisms exhibit enzymatic and priming response to root mucilage under drought, Soil Biology and Biochemistry, 116, 410–418, https://doi.org/10.1016/j.soilbio.2017.10.041, 2018.

Alvarez-Clare, S., Mack, M. C., and Brooks, M.: A direct test of nitrogen and phosphorus limitation to net primary productivity in a lowland tropical wet forest, Ecology, 94, 1540–1551, https://doi.org/10.1890/12-2128.1, 2013.

Becknell, J. M., Vargas G., G., Pérez-Aviles, D., Medvigy, D., and Powers, J. S.: Above-ground net primary productivity in regenerating seasonally dry tropical forest: Contributions of rainfall, forest age and soil, Journal of Ecology, 109, 3903–3915, https://doi.org/10.1111/1365-2745.13767, 2021.

Cunha, H. F. V., Andersen, K. M., Lugli, L. F., Santana, F. D., Aleixo, I. F., Moraes, A. M., Garcia, S., Di Ponzio, R., Mendoza, E. O., Brum, B., Rosa, J. S., Cordeiro, A. L., Portela, B. T. T., Ribeiro, G., Coelho, S. D., de Souza, S. T., Silva, L. S., Antonieto, F., Pires, M., Salomão, A. C., Miron, A. C., de Assis, R. L., Domingues, T. F., Aragão, L. E. O. C., Meir, P., Camargo, J. L., Manzi, A. O., Nagy, L., Mercado, L. M., Hartley, I. P., and Quesada, C. A.: Direct evidence for phosphorus limitation on Amazon forest productivity, Nature, 608, 558–562, https://doi.org/10.1038/s41586-022-05085-2, 2022.

Kavanagh, T. and Kellman, M.: Seasonal Pattern of Fine Root Proliferation in a Tropical Dry Forest, Biotropica, 24, 157, https://doi.org/10.2307/2388669, 1992.

Kummerow, J., Castillanos, J., Maas, M., and Larigauderie, A.: Production of fine roots and the seasonality of their growth in a Mexican deciduous dry forest, Vegetatio, 90, 73–80, https://doi.org/10.1007/BF00045590, 1990.

Lambers, H., Chapin, F. S., and Pons, T. L.: Photosynthesis, in: Plant Physiological Ecology, edited by: Lambers, H., Chapin, F. S., and Pons, T. L., Springer, New York, NY, 11–99, https://doi.org/10.1007/978-0-387-78341-3_2, 2008.

Lu, X., Vitousek, P. M., Mao, Q., Gilliam, F. S., Luo, Y., Zhou, G., Zou, X., Bai, E., Scanlon, T. M., Hou, E., and Mo, J.: Plant acclimation to long-term high nitrogen deposition in an N-rich tropical forest, PNAS, 115, 5187–5192, https://doi.org/10.1073/pnas.1720777115, 2018.

Muff, S., Nilsen, E. B., O'Hara, R. B., and Nater, C. R.: Rewriting results sections in the language of evidence, Trends in Ecology & Evolution, 37, 203–210, https://doi.org/10.1016/j.tree.2021.10.009, 2022.

Reid, J. P., Schnitzer, S. A., and Powers, J. S.: Short and Long-Term Soil Moisture Effects of Liana Removal in a Seasonally Moist Tropical Forest, PLOS ONE, 10, e0141891, https://doi.org/10.1371/journal.pone.0141891, 2015.

Schwartz, N. B., Medvigy, D., Tijerin, J., Pérez-Aviles, D., Rivera-Polanco, D., Pereira, D., Vargas G., G., Werden, L., Du, D., Arnold, L., and Powers, J. S.: Intra-annual variation in microclimatic conditions in relation to vegetation type and structure in two tropical dry forests undergoing secondary succession, Forest Ecology and Management, 511, 120132, https://doi.org/10.1016/j.foreco.2022.120132, 2022.

Wang, R., Goll, D., Balkanski, Y., Hauglustaine, D., Boucher, O., Ciais, P., Janssens, I., Penuelas, J., Guenet, B., Sardans, J., Bopp, L., Vuichard, N., Zhou, F., Li, B., Piao, S., Peng, S., Huang, Y., and Tao, S.: Global forest carbon uptake due to nitrogen and phosphorus deposition from 1850 to 2100, Global Change Biology, 23, 4854–4872, https://doi.org/10.1111/gcb.13766, 2017.

Waring, B. G., Pérez-Aviles, D., Murray, J. G., and Powers, J. S.: Plant community responses to stand-level nutrient fertilization in a secondary tropical dry forest, Ecology, 100, e02691, https://doi.org/10.1002/ecy.2691, 2019.

Wright, S. J., Yavitt, J. B., Wurzburger, N., Turner, B. L., Tanner, E. V. J., Sayer, E. J., Santiago, L. S., Kaspari, M., Hedin, L. O., Harms, K. E., Garcia, M. N., and Corre, M. D.: Potassium, phosphorus, or nitrogen limit root allocation, tree growth, or litter production in a lowland tropical forest, Ecology, 92, 1616–1625, https://doi.org/10.1890/10-1558.1, 2011.

---

## Referee Report (RR1)

The authors have done an admirable job creating a cohesive unit from their multiple response variables. This is a strong dataset which hints at some interesting trade-offs between nutrient and water limitation in tropical dry forests. With one exception, I feel the authorship team has done a sufficient job addressing my earlier concerns. The outstanding issue is the statistical justification for including plant-functional traits and canopy position as an explanatory variable in their uneven design. While this is well explained in the response-to-reviewers, much of this context is still missing from the manuscript itself (see line notes below).

Line Notes

68 Is the reference to water use efficiency as a trade-off with photosynthetic efficiency needed here? Could be an opportunity to simplify.

75 replace tropical with TDF

119 remove 'on'

Paragraph starting on 155 - Seems dishonest to call it fully factorial unless this is qualified along the lines of - "The nutrient-addition and drought aspects of the experiment were fully factorial…"

226 & 235 change productivity to "production"

267 How was substrate-derived microbial biomass C distinguished from other biomass?

306 How does PFT and understory/plantation fit into stand here?

401 Not clear how PFT and understory/plantation tree were analyzed, not currently included in the statistical analysis section.

403 This is a fair and clearly stated summary of the results, well done!

428 Add, "although these effects were not statistically significant"

436 Occam's razor would argue that effects weren't found because the manipulation was not strong enough.

440 Add, p = 0.09

444 Not sure of the grammar rules for botanical authorities, possible that A. Rich should be within ()

452 Similar to my comment for lines 306 and 401 above, there is no explanation of how this effect was tested. Regardless, the claim seems so weak as to not be worth mentioning?

469 Results sections says that F had one nodule, and D+F had 57, please revise.

482 Remind the reader that this is a glucose-based measure of CUE

---

## Author Response (AR2)

THE UNIVERSITY OF UTAH ®

School of Biological Sciences
South Biology Building
257 S. 1400 E., Salt Lake City, Utah 84112

14th March 2023
german.vargas@utah.edu

Dear Dr. Bahn,

We thank you for the opportunity to resubmit our manuscript bg-2022-203 *"Throughfall exclusion and fertilization effects on tropical dry forest tree plantations, a large-scale experiment"*. We also want to thank the reviewers and the Associate Editor Dr. Richard Nair for highlighting the areas in which there was room for improvement in the text. In the revised version you will find the following changes:
- Added clarity in the description of the analysis where we test the effects of the experimental treatments on tree diameter relative growth rates.
- Incorporation of the technical corrections suggested by the referees.

Below we detail the changes that we made to the manuscript in relation to these points. For clarity, the reviewers' suggestions appear in plain text, and our responses appear in blue text. If there is any additional way that I can facilitate the publication of our manuscript, do not hesitate to contact me. Thank you for your attention.

Sincerely,

German Vargas Gutiérrez, Ph.D.
NOAA C&GC Postdoctoral Fellow
School of Biological Sciences
The University of Utah
Salt Lake City, UT 84112

**Referee: 1**

Dear authors, I am very satisfied with your responses to my questions and concerns and also with the improved text. I believe that adding clarity in some parts of the manuscript was crucial to improve understanding and increase even further its quality. These data are of great value to the tropical ecology community and I really appreciate the effort of the authors in gathering years of data and telling now a nice story, strengthening the ecological implications of both nutrients and water shaping tropical dry forests. I do have very minor technical corrections that I list below and I also want to mention that in the response letter, mentions to figure/table numbers and lines were not always right so make sure to double check this for the resubmission to ease the review process.

We really appreciate, once again, the constructive feedback provided. We incorporated the suggested technical corrections in the new revised version of the manuscript. Also, to avoid issues with page numbering we indicate the line number of the changes for both the final document and the track changes document.

Technical corrections suggested:

- Line 25: Be consistent in using either throughfall or through-fall.

We replaced through-fall with throughfall in line 25 as that was a product of automatic correction in the text editor. We apologize for the confusion.

- Line 26: Sometimes control is referred to as CN, sometimes as C (for example Table S2).

To be consistent with the figures, we now refer to control plots as "control" in both the supplements and the main text. We deleted the acronym CN from the following lines:

Final document (26, 155), track changes document (26, 168).

- Line 119: "measurements of on fine roots" perhaps the "on" should be deleted.

We removed "on" as it was left out from the previous draft by mistake. Now the text reads:

Lines: final document (119), track changes document (131).

"…measurements of fine roots production…"

- Line 848: In table 1 I would suggest to add the meaning of each leaf habit (LH) also to the table caption.

We modified the table legend to improve clarity. Now the text reads:

Lines: final document (867), track changes document (918).

"…Here we present species leaf habit (LH) as deciduous (DC), semi-deciduous (SD), or evergreen (EV), whether the species is nitrogen fixer (NF), specific leaf area …"

- Line 189: I am assuming that these measurements were also performed from 2016 to 2020, following the description in line 181 for tree growth. It might be worth, however, to add this information to either description in line 124 (not only the experiment, meaning drought and

fertilisation happened for 4 years, but also the measurement?) or to each method description in case the dates are different depending on measurements.

We apologize for the confusion here. We added the months and years associated with each measurement in the text. In the example noted by the referee, the text now reads:

Lines: final document (190), track changes document (206).

"…We measured canopy productivity from January 2017 through December 2020 using two…"

- Line 225: Is there a diameter definition for "fine" root productivity or all roots inside the ingrowth core were considered?

We acknowledge the importance of defining root orders and root diameter, particularly in studying root ecology and physiology. In our case, we were only focused on the production of roots within the ingrowth core. Therefore, all the roots within the core were considered We apologize for the confusion here. We added a sentence that briefly describes what we consider fine roots:

Lines: final document (227-229), track changes document (243-245).

"…of 15 cm. With this method, we quantified fine roots as the biomass of new root growth inside an 8 cm diameter cylindrical ingrowth bag with a 2 mm nylon mesh. The cores…"

**Referee: 2**

The authors have done an admirable job creating a cohesive unit from their multiple response variables. This is a strong dataset which hints at some interesting trade-offs between nutrient and water limitation in tropical dry forests. With one exception, I feel the authorship team has done a sufficient job addressing my earlier concerns. The outstanding issue is the statistical justification for including plant- functional traits and canopy position as an explanatory variable in their uneven design. While this is well explained in the response-to-reviewers, much of this context is still missing from the manuscript itself (see line notes below).

We are grateful to the referee for the constructive feedback provided and the time invested in reviewing our manuscript. In this new version we try to make clear the details of the analysis that were absent in the text, we apologize for the lack of clarity.

Line Notes

- Line 68: Is the reference to water use efficiency as a trade-off with photosynthetic efficiency needed here? Could be an opportunity to simplify.

We thank the referee for pointing this out as there is also redundancy with another sentence in the paragraph. We proceeded to simplify the text by merging those two sentences as follows:

Lines: final document (67-69), track changes document (69-71).

"…In low-nutrient environments, plants maximize transpiration rates to increase mass flow nutrient uptake, but variations in water availability could limit these processes with potential costs to ecosystem productivity…"

- Line 75 replace tropical with TDF

We replaced tropical with TDF. Now the text reads:

Lines: final document (73), track changes document (84).

"…it is well documented that TDF leaf phenological cycles…"

- Line 119 remove 'on'

We removed "on" as it was left out from the previous draft by mistake. Now the text reads:

Lines: final document (119), track changes document (131).

"…measurements of fine roots production…"

- Paragraph starting on 155: Seems dishonest to call it fully factorial unless this is qualified along the lines of - "The nutrient-addition and drought aspects of the experiment were fully factorial..."

We apologize if the text was misleading. Indeed, we refer here to the experimental manipulations that we performed during the four years of our study. Whereas species identity, functional type, stand, and leaf habit, were factors that we did not manipulate as they were inherent to the plantation system in which we performed the experiment. We modified the text to make this clear.

Lines: final document (154-155), track changes document (167-168).

"…We performed nutrient and drought manipulations using a fully factorial design with four treatments: fertilization (F), drought (D), drought+fertilization (D+F), and un-manipulated control…"

- Lines 226 & 235: change productivity to "production"

We updated the text accordingly.

- Line 267: How was substrate-derived microbial biomass C distinguished from other biomass?

We apologize for the lack of detail. We now updated the text to make it clear to the reader and cited Kane et al. (2023) where the method is described in greater detail.

Lines: final document (268-270), track changes document (290-292)

"…, where we distinguished substrate-derived microbial biomass C from total microbial biomass by using the atom% $^{13}$C to calculate the total amount of $^{13}$C-labeled biomass per gram of dry soil (Kane et al., 2023)…"

- Line 306: How does PFT and understory/plantation fit into stand here?

We acknowledge that in the last version of the text, the description of the statistical analysis in which we considered PFT and understory/plantation was not clear. We apologize for that. We proceeded to expand this section by adding lines differentiating the tree diameter relative growth analysis from the analysis used to study changes in biomass fluxes and LAI metrics. The text now reads:

Lines: final document (302-320), track changes document (324-344).

"…We tested the effects of the experimental treatments on aboveground and belowground ecosystem processes by fitting a series (one for each response variable) of a two-factorial linear mixed effects model. For tree diameter RGR, we studied responses by understory and plantation trees separately due to differences in the life history of individuals and the possible biases in growth associated with tree size (Iida et al., 2014). Moreover, in addition to the treatment effects, we quantified the effects of two plant functional type classifications. For this, we fitted a model that included leaf phenology (*e.g.*, deciduous and evergreen) and a model that included whether the species was a nitrogen-fixer or not. Functional types are linked to physiological differences among tree species (Vargas G. et al., 2021; Powers and Tiffin, 2010; Vargas G. et al., 2015), and are important drivers explaining tree growth responses to nutrient additions and water availability (Waring et al., 2019; Costa et al., 2010; Wright et al., 2011; Toro et al., 2022). In these models, RGR was the response variable, and drought, fertilizer, and functional type were the predictors. Additionally, we included the species' identity of each stem nested within the plot nested within stand as random effects. In the case of biomass fluxes, microbial CUE, and LAI-derived metrics, these processes (*e.g.*, total litterfall) were the response variables, the drought treatment was one factor, and the fertilizer treatment was the second factor, we included their interaction and the experimental unit (*e.g.*, litterfall basket) nested within the plot nested within stands as a random intercept. With these models, we were able to estimate the main effect of drought, the main effect of fertilization, and the interaction between drought and fertilization, while also accounting for the effects of the plantation stand and the plot, and in the case of RGR plant functional type. We then calculated type III sum squares and the F value for each model in an analysis of variance (ANOVA) given our unbalanced design and used Tukey's HSD test for multiple comparisons…"

- Line 401: Not clear how PFT and understory/plantation tree were analyzed, not currently included in the statistical analysis section.

See the answer to the previous comment about the same issue.

- Line 403: This is a fair and clearly stated summary of the results, well done!

Thank you!

- Line 428: Add, "although these effects were not statistically significant"

We apologize for the confusion. We were referring to the relative contribution of each biomass flux to NPP. We added the suggested line to make this clear,

Lines: final document (440-441), track changes document (473-474).

"…plots (Fig. S14). Although, the observed changes in woody and root production were not statistically significant when analyzed individually. Changes…"

- Line 436: Occam's razor would argue that effects weren't found because the manipulation was not strong enough.

We apologize for the lack of clarity. In the text, we tried to acknowledge that the mild manipulations of soil moisture did not trigger any changes in litterfall as has been observed in other similar studies. To make this more clear, we added some text at the end of the last sentence in the paragraph.

Lines: final document (449-452), track changes document (482-485).

"…Our results are comparable to other throughfall exclusion experiments in which fine litter production was not affected by the drought treatment (Nepstad et al., 2002; Brando et al., 2006; Schwendenmann et al., 2010),

with most of its variation linked to inter-annual climatic variability rather than the experimental manipulations (Brando et al., 2008).…"

- Line 440: Add, p = 0.09

We updated the text accordingly.

- Line 444: Not sure of the grammar rules for botanical authorities, possible that A. Rich should be within ()

For *Biogeosciences* this is not specified in the author's guidelines. In the binomial system both approaches "(A. Rich)" or "A. Rich" are accepted. While arbitrary, we decided to use the approach without parenthesis as is the most used in the literature we are familiar with.

- Line 452: Similar to my comment for lines 306 and 401 above, there is no explanation of how this effect was tested. Regardless, the claim seems so weak as to not be worth mentioning?

We do acknowledge that the effects are weak. However, the growth rates of understory plant community n-fixers like *Vachellia collinsii* or *Gliricida sepium* tend to be lower than those of non-n-fixers such as *Sterculia apetala*, *Guazuma ulmifolia* or *Cecropia peltata*. The reason for this is associated with their higher wood density (Powers and Tiffin, 2010). Therefore, we considered it important to try to understand why there were no differences in their understory growth rates.

Our changes regarding this comment are threefold:
1- We updated the methods section to include a description of this analysis.
2- We simplified the discussion to make the above explanation clear. In summary, we try to explain that more than experimental manipulations there was an increase in light availability which probably trigger these changes in growth rates in D+F plots.
3- We did not consider it important to explain why there are no differences between functional groups in plantation trees. The reason for this is that in tree plantations diameter growth usually reaches a plateau and a self-thinning process kicks in. This might be the cause of the observed mortality. At the same time, two hurricanes (2016-Otto and 2017-Nate) hit the region, and in other drought experiments mortality has been observed at the beginning probably due to the installation of the treatments or sudden changes in growing conditions that affected trees that might have been already weakened by pest or competition (Rowland et al., 2015; Costa et al., 2010; Meir et al., 2018).

We streamlined the text as:

Lines: final document (466-478), track changes document (505-517).

"No species showed significant changes in RGR, but the understory trees showed a reduction in the differences between N-fixing and non-N-fixing trees. For F and D this was due to a reduction in growth rates by non-N-fixing trees, while for D+F due to an increase in the growth rates by N-fixing trees (Fig. 4). One possible reason for these patterns could be increased resource availability due to decreased competition. The D+F plots in which these three species were present experienced the highest biomass losses due to mortality during the four years of experimental manipulation (Table S5; Fig. S10). Even though it is hard to determine the cause of death, an increase in growth rates of understory trees has been observed after the mortality of larger trees (Rowland et al., 2015). The lack of responsiveness in the F and D plots, in addition to the biomass losses in some of the D+F plots (Table S5), supports the idea that the availability of resources such as light could be the cause of higher RGR in the D+F compared to the other treatments (Fig. S15). The lowest RGR were found in plots with the D treatment, with the strongest experimental effect on *D. retusa*, *E. cyclocarpum*, and *S. glauca* (Fig. S9). Yet not

significant, these results are very similar to what has been found in other tropical throughfall exclusion experiments (Meir et al., 2015), in which there is an overall negative effect on tree diameter growth by a decrease in soil moisture."

- Line 469: Results sections says that F had one nodule, and D+F had 57, please revise.

We updated the text accordingly.

Lines: final document (480), track changes document (519).

- Line 482: Remind the reader that this is a glucose-based measure of CUE

We updated the text accordingly.

Lines: final document (493), track changes document (544).
* * *
**Editor's comments to the author(s):**

A small revision is still required to add a little more explanation of the interpretation of the species effects in the paper.

Additional private note (visible to authors and reviewers only):
I agree with R2 that a short justification of how the unbalanced species design is interpreted in the statistical analysis would be useful in the paper.
An attentive reader would notice that the effective sample size of these species combinations is 1 and yet there is a substantial part of the discussion about these species effects. While obviously working in the diverse tropical forests is challenging and this is a perhaps necessary limit of a valuable multifactorial experiment, this should be spelt out clearly in the text. This information is already in the response to reviewers and should be simple to quickly incorporate without any major change to the paper.

**Response to the editor:** We highly appreciate the comments by the Associate Editor highlighting the main areas of the text that needed clarification. In the present version, we incorporated the suggestions by the referees and highlighted that we studied diameter growth responses using functional types and not species. We also provided a justification in the analysis section of why we consider this approach.

**References:**

[revised manuscript text omitted]